# Fragile X mental retardation protein is a Zika virus restriction factor that is antagonized by subgenomic flaviviral RNA

Ruben Soto-Acosta[1], Xuping Xie[1], Chao Shan[1], Coleman K Baker[2], Pei-Yong Shi[1,3,4,5,6], Shannan L Rossi[2,6,7,8], Mariano A Garcia-Blanco[1,6,9]*, Shelton Bradrick[1,6,8]*

[1]Department of Biochemistry and Molecular Biology, University of Texas Medical Branch, Galveston, United States; [2]Department of Microbiology and Immunology, University of Texas Medical Branch, Galveston, United States; [3]Sealy Center for Vaccine Development, University of Texas Medical Branch, Galveston, United States; [4]Department of Pharmacology & Toxicology, University of Texas Medical Branch, Galveston, United States; [5]Sealy Center for Structural Biology & Molecular Biophysics, University of Texas Medical Branch, Galveston, United States; [6]Institute for Human Infections & Immunity, University of Texas Medical Branch, Galveston, United States; [7]Department of Pathology, University of Texas Medical Branch, Galveston, United States; [8]Center for Biodefense & Emerging Infectious Diseases, University of Texas Medical Branch, Galveston, United States; [9]Programme in Emerging Infectious Diseases, Duke-NUS Medical School, Singapore, Singapore

**Competing interests:** The authors declare that no competing interests exist.

**Abstract** Subgenomic flaviviral RNA (sfRNA) accumulates during infection due to incomplete degradation of viral genomes and interacts with cellular proteins to promote infection. Here we identify host proteins that bind the Zika virus (ZIKV) sfRNA. We identified fragile X mental retardation protein (FMRP) as a ZIKV sfRNA-binding protein and confirmed this interaction in cultured cells and mouse testes. Depletion of FMRP elevated viral translation and enhanced ZIKV infection, indicating that FMRP is a ZIKV restriction factor. We further observed that an attenuated ZIKV strain compromised for sfRNA production was disproportionately stimulated by FMRP knockdown, suggesting that ZIKV sfRNA antagonizes FMRP activity. Importantly, ZIKV infection and expression of ZIKV sfRNA upregulated endogenous FMRP target genes in cell culture and ZIKV-infected mice. Together, our observations identify FMRP as a ZIKV restriction factor whose activity is antagonized by the sfRNA. Interaction between ZIKV and FMRP has significant implications for the pathogenesis of ZIKV infections.
DOI: https://doi.org/10.7554/eLife.39023.001

## Introduction

Zika virus (ZIKV) is a mosquito-borne flavivirus that is closely related to dengue (DENV) and yellow fever viruses (YFV). There are three ZIKV genotypes: one from Asia, which has caused the recent pandemic in the Americas, and two from Africa (East and West African) (*Lanciotti et al., 2016*). ZIKV infection is characterized by fever, arthralgia and conjunctivitis (*Goeijenbier et al., 2016*) and was considered an exceedingly rare human infection since only 14 cases were reported prior to 2007 (*Faye et al., 2014*). In 2007, the first ZIKV outbreak caused by an Asian lineage was reported in

**eLife digest** Certain mosquitoes can carry pathogens that are able to infect humans, including Zika and dengue viruses. Most people infected with Zika virus only develop mild symptoms, or no symptoms at all. But if the virus infects a pregnant woman, it can lead to miscarriage and other pregnancy complications, or cause severe birth defects in her unborn baby.

Viruses must infect the cells of a host to multiply. To do so, they hijack the cellular machinery to make proteins needed to copy their genetic material and assemble new virus particles. The genetic material of Zika virus is made of ribonucleic acid (RNA). When the Zika virus infects cells, pieces of the virus RNA, known as subgenomic flavivirus RNAs (or sfRNAs for short), accumulate in the cell.

Cells infected with dengue virus, which is closely related to the Zika virus, also accumulate sfRNA. Dengue sfRNA is known to bind to and inhibit the activity of specific proteins in cells that would otherwise block the virus from multiplying. Nonetheless, it is not clear whether the sfRNA from Zika virus performs a similar role.

Soto-Acosta et al. searched for human proteins that could bind to Zika sfRNA and may affect the ability of the virus to multiply. The experiments showed that a protein known as FMRP, which, when faulty, is linked to a genetic condition that causes a range of developmental problems, binds to Zika sfRNA in human and mouse cells infected with Zika virus. FMRP inhibits the production of virus proteins in the cells and limits the ability of the virus to multiply. However, as Zika sfRNA gradually accumulates during infection, the sfRNA binds to FMRP and interferes with its activity, allowing the virus to multiply more efficiently. Soto-Acosta et al. also found that Zika sfRNA affects the ability of FMRP to regulate the production of other proteins that are normally found in cells.

These findings suggest that the interference of the virus with FMRP may contribute to Zika disease in humans. Moreover, a mutant Zika virus unable to produce sfRNA could be developed into a vaccine to potentially prevent Zika.

DOI: https://doi.org/10.7554/eLife.39023.002

Micronesia, followed by epidemics in French Polynesia in 2013 and the recent pandemic (2015–2016) in the Americas (*Weaver et al., 2016*; *Aliota et al., 2017*). During this period, it has been estimated that 1.5 million cases occurred in Brazil and more than 25,000 cases in Colombia (*Focosi et al., 2016*; *Samarasekera and Triunfol, 2016*). As of January 2018, the virus was widely distributed in 50 countries in the Americas (*PAHO/WHO, 2018*). The public health concern about Zika has been primarily driven by its maternal-fetal (*Brasil et al., 2016a*; *Yockey et al., 2016*; *Driggers et al., 2016*) and sexual modes of transmission (*Deckard et al., 2016*; *Davidson et al., 2016*; *Hills et al., 2016*) as well as its association with congenital abnormalities, especially microcephaly, and Guillain-Barré syndrome in adults (*Krauer et al., 2017*; *Martines et al., 2016*; *Miner et al., 2016*; *Brasil et al., 2016b*). These unique aspects set ZIKV apart from other flavivirus infections and have spurred efforts to understand ZIKV host-pathogen interactions.

The ZIKV life-cycle starts with virus attachment and receptor-mediated endocytosis (*Hamel et al., 2015*; *Meertens et al., 2017*). Upon endosome acidification, the viral envelope fuses with the endosomal membrane, releasing the nucleocapsid into the cytoplasm for uncoating and initial viral translation at the cytosolic surface of the endoplasmic reticulum (ER). Translation produces a single polyprotein that is cleaved co- and post-translationally by cellular and viral proteases into 10 mature proteins: three structural proteins forming the virion (capsid, C; pre-membrane, prM; and envelope, E) and seven non-structural proteins (NS1, NS2A, NS2B, NS3, NS4A, NS4B and NS5) required for viral replication and inhibition of host defense mechanisms (*Campos et al., 2017*; *Barrows et al., 2018*). Replication of ZIKV RNA and assembly of viral particles occur in close association with rearranged ER membranes (*Rossignol et al., 2017*). Assembled virions are transported through the secretory pathway, where the furin protease cleaves prM into and M, resulting in mature virions that are secreted into the extracellular space (*Hasan et al., 2018*).

The flaviviral genome contains 5′ and 3′ untranslated regions (UTR) that are essential for genome cyclization and initiation of RNA synthesis (*Filomatori et al., 2006*; *Alvarez et al., 2005*). The ZIKV 3′ UTR is highly structured and consists of four domains: xrRNA1, xrRNA2, the dumbbell (DB) and the 3′ SL (*Zhu et al., 2016*). The ZIKV 3′ UTR contains confirmed and predicted pseudoknot

interactions located in xrRNA1 and xrRNA2, respectively. These pseudoknots are important for stalling of the cellular 5′ to 3′ exonuclease, XRN1, and accumulation of at least two sfRNA species (sfRNA1 and sfRNA2) (*Akiyama et al., 2016*). The sfRNA of a few different flaviviruses has been described to exert pro-viral functions, possibly through acting as a 'sponge' for antiviral host proteins (*Göertz et al., 2018*). The DENV sfRNA has been shown to interact with stress granule associated proteins G3BP1, G3BP2 and CAPRIN1 and the ubiquitin ligase TRIM25 to dampen innate immune responses (*Bidet et al., 2014*; *Manokaran et al., 2015*). Similarly, Moon et al reported that DENV-2 and KUNV sfRNAs inhibited XRN1 activity, leading to accumulation of uncapped transcripts in infected cells and disruption of mRNA homeostasis that could potentially deregulate antiviral responses (*Moon et al., 2012*). Regarding ZIKV, Musashi-1 (MSI1) was reported to be required for efficient infection and interact with viral genomes; additionally, ZIKV infection disrupted the activity of MSI1 (33). However, whether or not ZIKV sfRNA interacts with MSI1 and modulates its activity is unclear.

In this work, we identified the Fragile X Mental Retardation Protein (FMRP) as a host factor that binds to the ZIKV sfRNA and to viral genomes. Functional assays indicated that FMRP represses ZIKV infection by inhibiting viral translation. Analysis of an attenuated ZIKV vaccine candidate (Δ10 ZIKV) that is defective for sfRNA production suggests that sfRNA enhances ZIKV infection partly through antagonizing FMRP activity. Additionally, we observed that ZIKV infection blocks the canonical activity of FMRP and increases the expression of FMRP target genes. Finally, we present evidence implicating deregulation of FMRP activity in a mouse model of ZIKV infection, suggesting that ZIKV pathogenesis may involve virus-mediated FMRP inhibition.

## Results

### FMRP interacts with the ZIKV 3′ UTR in cell extracts, infected cells and infected mouse testes

Before this work, there was no knowledge of host proteins that bind and are modulated by ZIKV sfRNA. We previously used RNA affinity chromatography coupled with quantitative mass spectrometry (RAC-MS) to identify proteins that interact with DENV-2 sfRNA (*Bidet et al., 2014*; *Manokaran et al., 2015*). Here, we used a similar approach with lysates from JEG3 (choriocarcinoma) cells, which are permissive for ZIKV infection, to identify host proteins that interact with full-length ZIKV 3′ UTRs of the pandemic Asian lineage strain PRVABC59 and the African strain MR766. This analysis revealed 27 proteins that preferentially interacted with the ZIKV RNAs compared to an RNA derived from the coding sequence of DENV (*Table 1*-source data 1). We also analyzed a second negative control RNA derived from vector backbone sequence and obtained similar results (data not shown). We did not detect major differences in the protein-binding profiles of PRVABC59 and MR766 ZIKV sequences. Eight proteins that were identified by ≥2 unique peptides and were ≥2 times as abundant in both the ZIKV sequences relative to the control RNA were considered as high-confidence ZIKV 3′ UTR binding proteins (*Table 1*). The proteins showing highest preferential binding to ZIKV sfRNAs were FMRP and its two paralogs, Fragile X Related proteins 1 and 2 (FXR1 and FXR2) for both ZIKV 3′ UTRs.

In order to map the region(s) of the ZIKV 3′ UTR that mediate binding of FXR proteins, we performed an RNA affinity chromatography experiment using HeLa lysate and fragments of the ZIKV-PRVABC59 3′ UTR, and probed for proteins by western blotting (WB). As expected, the complete 3′ UTR interacted with FMRP, FXR1 and FXR2 (*Figure 1*, lane 1). Deletion of xrRNA1 dramatically reduced interaction with all three proteins (lane 2) and deletion of xrRNA1 and 2 eliminated detectable binding. Additionally, xrRNA1 alone was sufficient to bind FMRP, FXR1 and FXR2 (lane 4). These data indicate that xrRNA1 is necessary and sufficient for efficient binding to these proteins. We also probed for DDX6, G3BP1 and PTB. DDX6 interacted specifically with the ZIKV dumbbell region in contrast to FMRP. Interestingly, PTB and G3BP1, which were previously reported to interact with the DENV 3′ UTR (*Bidet et al., 2014*; *De Nova-Ocampo et al., 2002*), did not strongly interact with ZIKV 3′ UTR (*Figure 1*).

We chose to focus on FMRP due to its strong association with ZIKV RNA in binding experiments (*Figure 1B*), well-established association with human neurodevelopmental disease, and high level of expression in tissues relevant to ZIKV, such as brain and reproductive tract (*Devys et al., 1993*). To

**Table 1.** List of ZIKV 3′ UTR binding proteins identified by label-free mass spectrometry.

| Accession[*] | Unique peptides[†] | Confidence score[‡] | Normalized abundance[§] | | | Ratio vs NS2A[#] | |
|---|---|---|---|---|---|---|---|
| | | | NS2A | PRVABC59 3′ UTR | MR766 3′ UTR | PRVABC59 3′ UTR | MR766 3′ UTR |
| sp\|P51116\|FXR2_HUMAN | 2 | 129.6 | 1.8E + 04 | 6.1E + 05 | 8.2E + 05 | 34.3 | 46 |
| sp\|P51114\|FXR1_HUMAN | 12 | 897.8 | 6.2E + 05 | 8.2E + 06 | 9.9E + 06 | 13.4 | 16 |
| sp\|Q06787\|FMR1_HUMAN | 6 | 361.7 | 1.1E + 05 | 1.4E + 06 | 1.7E + 06 | 12.3 | 15 |
| sp\|P26196\|DDX6_HUMAN | 9 | 723.9 | 4.7E + 05 | 2.0E + 06 | 3.9E + 06 | 4.2 | 8 |
| sp\|Q9Y520\|PRC2C_HUMAN | 4 | 218.0 | 3.8E + 04 | 1.3E + 05 | 1.8E + 05 | 3.3 | 5 |
| sp\|P15927\|RFA2_HUMAN | 2 | 165.8 | 3.5E + 04 | 1.6E + 05 | 1.3E + 05 | 4.7 | 4 |
| sp\|Q8N0V3\|RBFA_HUMAN | 2 | 136.3 | 7.3E + 04 | 1.6E + 05 | 2.2E + 05 | 2.2 | 3 |
| sp\|P06748\|NPM_HUMAN | 5 | 623.9 | 5.5E + 06 | 1.1E + 07 | 1.6E + 07 | 2.0 | 3 |

*Unique identifier number of the protein.

†Number of peptide sequences uniquely identified for each protein.

‡Confidence score ($-10\log P$) reflects how well the MS/MS spectrum matches the peptides for all observed mass spectra that correspond to sequences within the protein. A higher score indicates a more confident match.

§Protein abundances were calculated by measuring the area under the curve of the corresponding peaks in the ion chromatogram.

#Protein abundances of RBPs interacting with PRVABC59 and MR766 ZIKV 3′ UTRs were compared with NS2A control RNA to calculate ratio of enrichment (PRVABC59 3′ UTR/NS2A and MR766 3′ UTR/NS2A). Proteins enriched >2 fold and with unique peptides $\geq$ 2 were considered to be high-confidence ZIKV 3′ UTR interacting proteins.

DOI: https://doi.org/10.7554/eLife.39023.003

The following source data is available for Table 1:

Source data 1.

DOI: https://doi.org/10.7554/eLife.39023.004

validate the interaction of FMRP with ZIKV 3′ UTR in the context of ZIKV infection, we performed RNA immunoprecipitation (RIP) with anti-FMRP antibody or isotype control and detected ZIKV RNAs using northern blot (NB) (*Figure 2A*). HeLa cells were infected with Dakar 41525 (ZIKV-Dakar), Cambodia FSS13025 (ZIKV-Cambodia) or PRVABC59 (ZIKV-Puerto Rico) strains, cell lysates were collected and FMRP was immunoprecipitated. Total RNA interacting with FMRP was isolated and the presence of ZIKV RNAs was detected by NB using a DNA probe that hybridizes to the ZIKV 3′ UTR. Both ZIKV sfRNA and viral genome co-precipitated with anti-FMRP but not control IgG (*Figure 2A*) for all three virus strains. Interestingly, quantification of sfRNA and genomic RNA signals indicated that FMRP preferentially interacts with the ZIKV sfRNA (*Figure 2B*). We additionally performed RIP with anti-PTB and isotype control antibodies and showed that viral genomes and sfRNA minimally co-precipitated with PTB and enrichment was much weaker than that for FMRP RIP (*Figure 2C*). We also analyzed the integrity of unbound viral RNAs present in supernatants after RIP. This analysis revealed that viral RNAs did not degrade during the RIP procedure (*Figure 2C*).We determined whether or not the interaction of FMRP with ZIKV RNA occurs in testes of infected mice. We selected testes for analysis because this tissue exhibits relatively high viral load in the A129 mouse model (IFNAR1 knockout) of ZIKV infection (*Dowall et al., 2016*; *Rossi et al., 2016*). FMRP was subjected to IP from testes lysates of mice infected with ZIKV-Cambodia for 6 days and RT-qPCR was used to measure co-precipitating viral RNA using qPCR assays targeting the viral genome exclusively or both the genome and sfRNA (*Figure 2D*). We observed enrichment of viral genomes by ~2 fold for FMRP IP compared to the negative control. Importantly, the enrichment was nearly twice as robust when using a qPCR assay that detects both genomes and the sfRNA. These results validate the interaction of FMRP with ZIKV RNA in vivo and, as observed in infected cultured cells, indicate preferential binding of FMRP to the ZIKV sfRNA.

## FMRP represses ZIKV infection by blocking viral RNA translation

Given the binding of FMRP to ZIKV sfRNA, we hypothesized that this protein would play an antiviral role for ZIKV. To analyze the functional relevance of FMRP in ZIKV infection, we infected HeLa cells

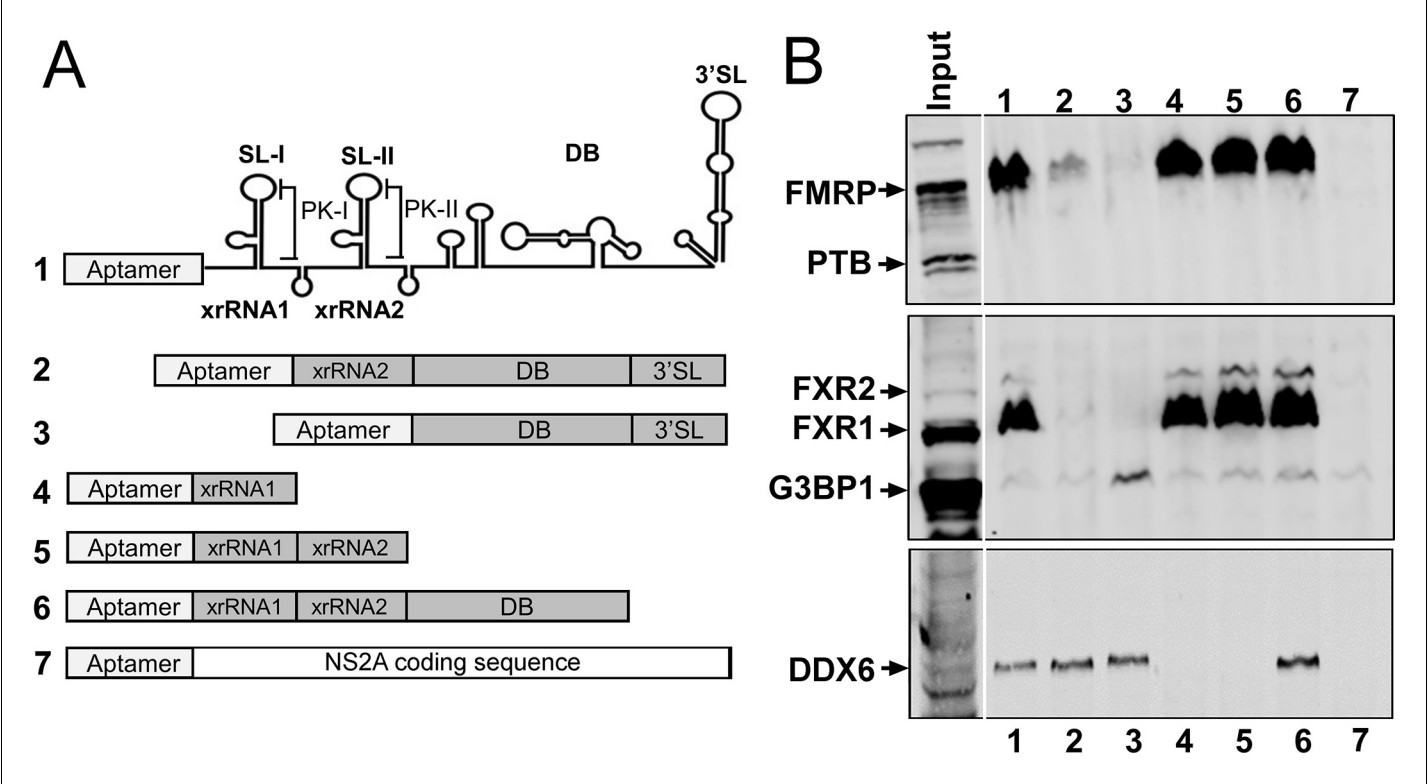

**Figure 1.** xrRNA1 is required and sufficient for FMRP binding to the ZIKV 3′ UTR. (**A**) Schematic illustrating the secondary structures present in the ZIKV 3′ UTR and RNAs used for affinity chromatography. Deletion mutant RNA constructs fused to a tobramycin aptamer at the 5′ end are shown. DENV NS2A coding sequence was used as a negative control RNA. (**B**) Purified RNAs bound to tobramycin-sepharose beads were incubated with HeLa cell lysate and unbound proteins were washed away prior to elution for western blotting for DDX6, FMRP, FXR1, FXR2, G3BP1 and PTB.
DOI: https://doi.org/10.7554/eLife.39023.005

in which FMRP was knocked down (KD) with individual siRNAs or a pool of four different siRNAs (*Figure 3A*). Forty-eight hours post-infection with ZIKV-Dakar at low MOI (0.01), we measured viral titers using plaque assay and rate of infection by flow cytometry. Both viral titers and rate of infection were increased by 2- to 5-fold due to FMRP KD (*Figure 3B and C*, *Figure 3—figure supplement 1A*). The pooled siRNAs effectively reduced FMRP levels and this correlated with large increases in ZIKV infection. The antiviral effect of FMRP was corroborated by immunofluorescence (IF) of infected cells to detect viral antigen and analysis by high-content imaging (*Figure 3D*). Moreover, we observed that FMRP KD enhanced infection with Asian strains (ZIKV-Cambodia and ZIKV-Puerto Rico) to different extents, but did not impact infection by DENV-2 (*Figure 3—figure supplement 1B and C*). We additionally tested the effect of FXR1 and FXR2 knockdown on ZIKV infection. We observed a slight increase in virus infection rate for FXR1 depletion while FXR2 knockdown significantly reduced infection (*Figure 3—figure supplement 2*). Together, these observations suggest that FMRP acts as a specific restriction factor for multiple ZIKV strains.

We also tested the effect of FMRP KD under experimental conditions that interrogate virus infection at higher MOI and shorter infection time, and observed that FMRP KD increased infection rate from ∼50% to∼80% (*Figure 3—figure supplement 3A and B*). Analysis of fluorescence intensity (FI) histograms (*Figure 3—figure supplement 3C*) revealed the presence of two distinct populations of infected cells: one of low FI and a second of high FI for expression of viral envelope (E) protein. Cells transfected with non-targeting siRNA (blue histogram) were predominantly in the low-infected cell population. In contrast, cells transfected with siRNAs targeting FMRP (red histogram) were mostly in the highly-infected population. We calculated FI mode values, indicating the maximum FI peak for each condition and found that FMRP knockdown led to a ∼ 10 fold increase, indicating elevated levels of E protein expression (*Figure 3—figure supplement 3D*). WB analysis revealed increased

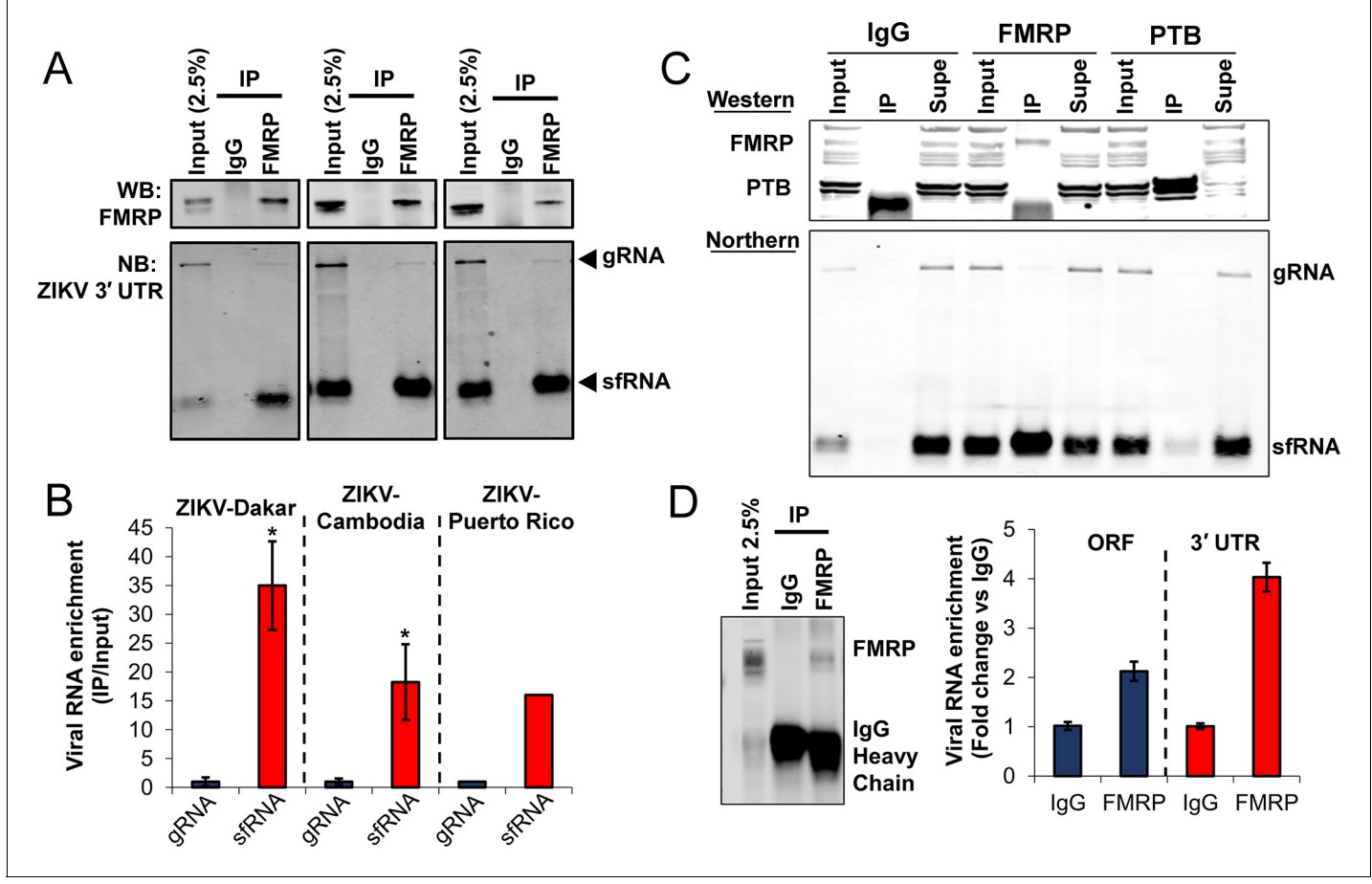

**Figure 2.** FMRP interacts with ZIKV RNA in infected cells. (**A**) RNA immunoprecipitation (RIP) was performed using either control IgG or anti-FMRP antibodies. HeLa cells were infected with the indicated virus at MOI of 3 and harvested 48 hr post-infection. Viral genome (gRNA) and the subgenomic flaviviral RNA (sfRNA) were detected by Northern blot (NB) using a DNA probe that hybridizes to the ZIKV 3' UTR. Western blotting (WB) shows the specificity of FMRP IP. (**B**) Bar graph showing the preferential binding of FMRP to ZIKV sfRNA. Densitometry analysis of NB assays was performed to determine relative levels of co-precipitating gRNA and sfRNA compared to input signals. Enrichment of sfRNA was normalized to the gRNA enrichment. Data represents the mean ± SEM of three independent experiments for ZIKV-Dakar and ZIKV-Cambodia (*p<0.05). One experiment was performed for ZIKV-Puerto Rico. (**C**) RNA IP from HeLa cells infected with ZIKV-Dakar using anti-PTB, anti-FMRP and isotype control antibodies. (**D**) FMRP interacts with ZIKV RNA in infected mouse testes. Testes from ZIKV-infected mice were lysed and used for FMRP-RIP. RT-qPCR assays were used targeting the ZIKV ORF, to measure viral genomes exclusively, and the 3' UTR, to measure both genomes and sfRNA. Bar graph represents the enrichment of viral RNAs in FMRP-pulldown normalized to the control IgG. Data indicates the mean ± range of two biological replicates. The WB shows IP of FMRP.

DOI: https://doi.org/10.7554/eLife.39023.006

The following source data is available for figure 2:

**Source data 1.**

DOI: https://doi.org/10.7554/eLife.39023.007

NS4B and NS2B protein levels in FMRP KD cells infected with ZIKV at MOIs of 1 and 5 (*Figure 3—figure supplement 3E and F*). Together, these data indicate that during a single ZIKV replication cycle, the absence of FMRP significantly increases both the rate of infection and the level of viral protein accumulation per cell.

Because FMRP is known to be a repressor of cellular mRNA translation (*Zalfa et al., 2003*; *Laggerbauer et al., 2001*), we hypothesized that translation of ZIKV is inhibited by FMRP early after infection. To test this hypothesis, we used an infectious ZIKV that expresses NanoLuc luciferase and analyzed early accumulation of NanoLuc in control and FMRP KD cells. Treatment with the translation elongation inhibitor cycloheximide (CHX) was used to control for background signal present in

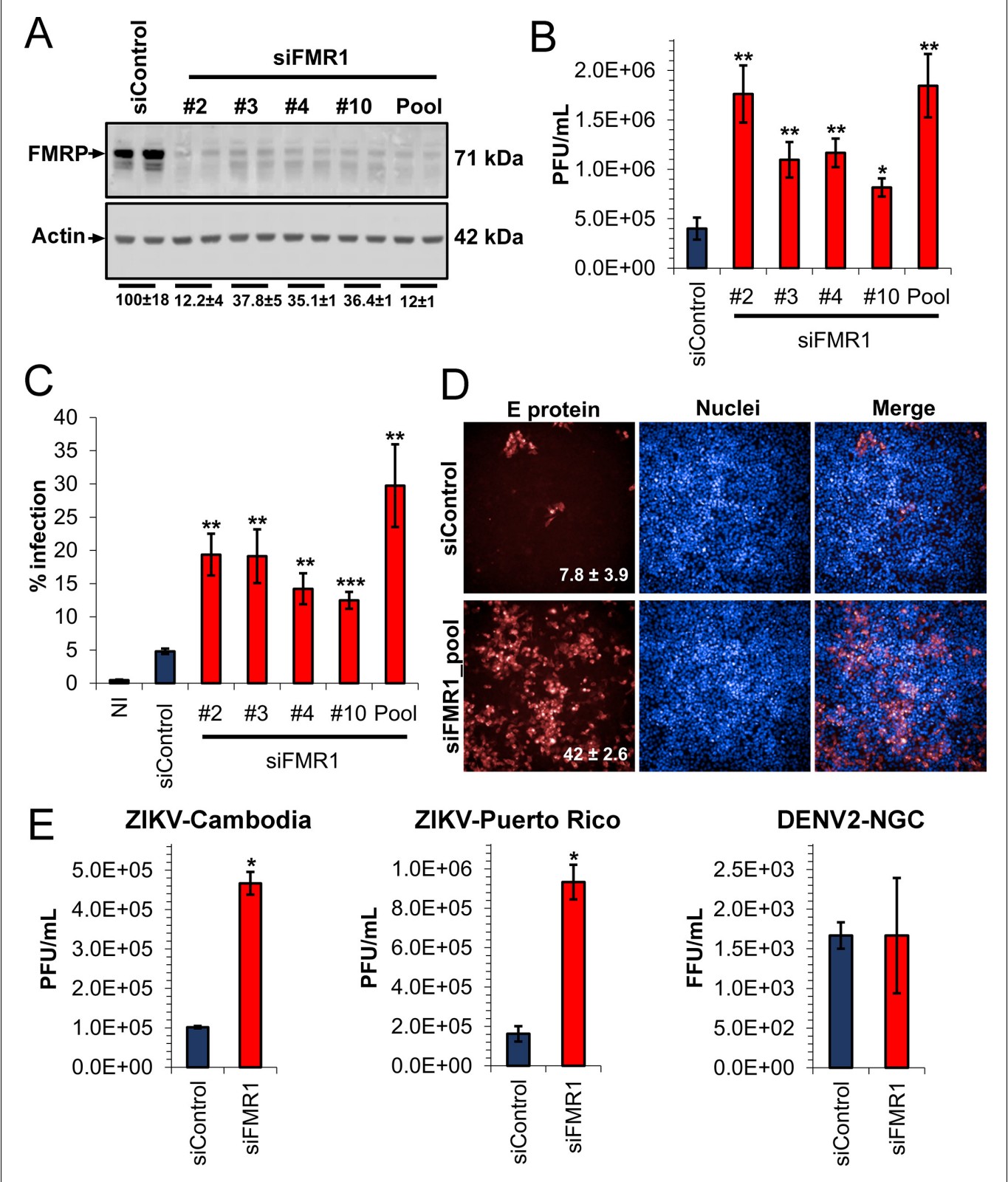

**Figure 3.** Depletion of FMRP increases ZIKV infection. HeLa cells were transfected with control siRNA (siControl) and individual or a pool of four different siRNAs targeting FMR1 (siFMR1). Cells were infected with ZIKV-Dakar at MOI 0.01 for 48 hr. (A) Representative WB of FMRP knockdown efficiency. Relative FMRP expression is indicated below. (B) Viral titers in supernatants were measured by plaque assay and are plotted as PFU/ml. (C) Infection rates were measured by immunofluorescence against viral envelope (E) protein using flow cytometry (representative scatter plots are shown in *Figure 3 continued on next page*

Figure 3 continued

*Figure 3—figure supplement 1A*) or (D) high-content imaging. Data represent the mean ±SEM of two independent experiments. (E) HeLa cells were transfected with siControl or siFMR1 (pool) and infected with ZIKV-Cambodia (MOI 0.01), ZIKV-Puerto Rico (MOI 0.01) or DENV-2-NGC (MOI 0.03) 48 hr later. At 48 hr post-infection, supernatants were collected and analyzed for viral titers. Graphs represent the mean ±SD of PFU/mL of one of three independent experiments for ZIKV-Cambodia and ZIKV-Puerto Rico. For DENV-NGC, three biological replicates are plotted (*p<0.05, **p<0.005, ***p<0.001). NI, non-infected cells.
DOI: https://doi.org/10.7554/eLife.39023.008
The following source data and figure supplements are available for figure 3:

**Source data 1.**
DOI: https://doi.org/10.7554/eLife.39023.012
**Figure supplement 1.** ZIKV Infection rates measured by flow cytometry.
DOI: https://doi.org/10.7554/eLife.39023.009
**Figure supplement 2.** Silencing of fragile X related proteins FXR1 and FXR2 has differential effect on ZIKV infection.
DOI: https://doi.org/10.7554/eLife.39023.010
**Figure supplement 3.** Depletion of FMRP increases ZIKV infection..
DOI: https://doi.org/10.7554/eLife.39023.011

the virus stock. The Nanoluc signal observed in control cells was well above that detected in the CHX-treated control at 3.5 hr post infection (hpi) indicating measurable viral protein synthesis at this early time point. Importantly, we observed a 4-fold increase in NanoLuc signal in FMRP depleted cells (siFMR1) compared to control siRNA transfected cells, suggesting that FMRP depletion results in increased of ZIKV translation (*Figure 4A and B*). Similar effects were observed in the presence of NITD008, a potent inhibitor of the flaviviral NS5 RNA-dependent RNA polymerase (*Deng et al., 2016*; *Yin et al., 2009*), ruling out a possible contribution of RNA replication to the increased luciferase signal in FMRP KD cells (*Figure 4A and B*). Consistent with this, we did not observe significant differences in viral RNA accumulation between control and FMRP KD cells in the absence or presence of NITD008 (*Figure 4C*). Finally, we calculated a 3.5-fold increase of ZIKV translation efficiency (NanoLuc levels normalized to ZIKV RNA) in FMRP KD cells compared to the negative control in both the absence and presence of NITD008 (*Figure 4D*). Together, these results strongly suggest that FMRP inhibits ZIKV infection by reducing viral translation.

## ZIKV sfRNA antagonizes FMRP function

Because FMRP binds preferentially to ZIKV sfRNA, we tested the hypothesis that sfRNA, which accumulates to high levels during the course of infection (*Figure 2A*), is capable of attenuating FMRP-mediated anti-ZIKV activity. For this purpose, we analyzed a ZIKV mutant (Δ10 ZIKV) that contains a 10-nt deletion within the 3′ UTR DB structure and is highly attenuated in mice and non-human primates (*Shan et al., 2017a*; *Shan et al., 2017b*). We first asked whether Δ10 ZIKV is deficient in sfRNA production. Analysis of viral RNA from infected HeLa cells demonstrated that Δ10 ZIKV has a defect in sfRNA accumulation. An sfRNA/gRNA ratio of 2.7 was observed in WT ZIKV infected cells but the ratio was only 0.6 in Δ10 ZIKV infected cells. These results suggest that Δ10 ZIKV is attenuated, as least in part, through loss of sfRNA accumulation caused by the 10-nt deletion (*Figure 5A and B*).

Next, we asked whether the WT and mutant viruses exhibit differential sensitivity to FMRP depletion. We observed that Δ10 ZIKV infection rate in HeLa cells is reduced compared to WT ZIKV under negative control conditions (~11% vs~25% infected cells) and, importantly, FMRP KD disproportionately enhanced Δ10 ZIKV infection rate (*Figure 5C*): whereas WT virus was enhanced by approximately 3-fold, the infection rate for Δ10 ZIKV rose by nearly 5-fold (*Figure 5D*). Representative scatter plots depicting the infected cell populations are shown in *Figure 5—figure supplement 1*. Additionally, similar effects were observed on levels of cell-associated ZIKV NS2B and NS4B proteins (*Figure 5E* to G). Together, these results suggest that sfRNA antagonizes FMRP and thus enhances ZIKV infection.

Given the key role of FMRP in neurodevelopment, its mechanism of action has been extensively studied and several cellular mRNAs are known targets for translational repression by FMRP (*Zalfa et al., 2003*; *Ascano et al., 2012*; *Korb et al., 2017*). We examined expression levels of ten reported targets in control and FMRP-depleted HeLa cells: *ARC RHOA, SOD1, RAC1, PNPLA6,*

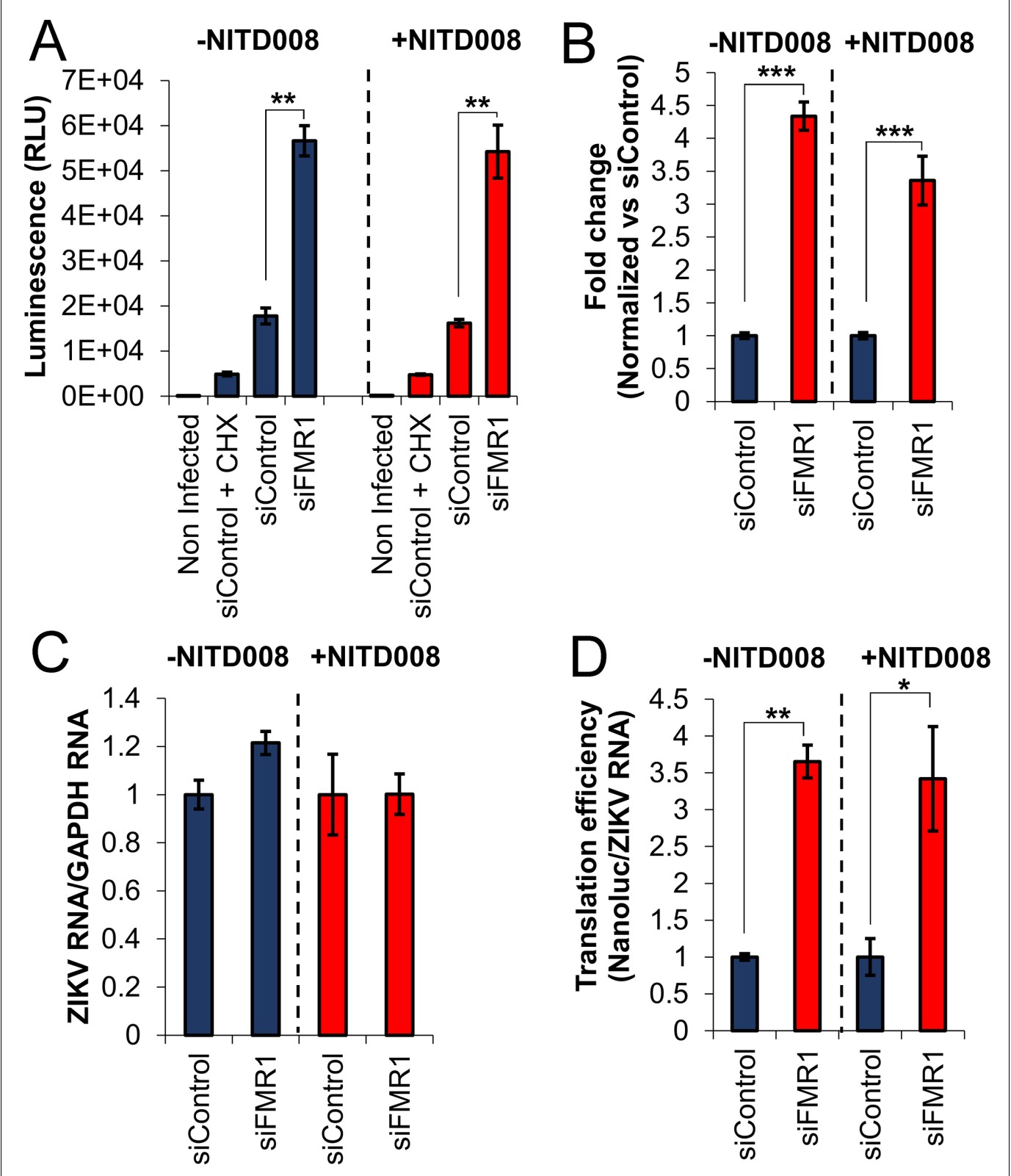

**Figure 4.** FMRP inhibits ZIKV translation early after infection. (**A**) An infectious ZIKV reporter virus expressing NanoLuc luciferase was used to evaluate ZIKV translation. HeLa cells were transfected with control siRNA (siControl) or the siRNA pool targeting the FMR1 gene (siFMR1). Two hours before infection, cells were pretreated with DMSO (-NITD008) or NITD008 at 20 µM. Accumulation of Nanoluc was evaluated at 3.5 hr post-infection. Cycloheximide (CHX) treatment was used to control for background signal present in the virus stock. (**B**) The graph shows luminescence signals with
*Figure 4 continued on next page*

*Figure 4 continued*

background (CHX control) subtracted and normalized to the siControl condition. (**C**) In parallel, viral RNA was measured by RT-qPCR, calibrated with GADPH mRNA expression and normalized to siControl. (**D**) Translation efficiency was calculated by normalizing the luminescence signals to ZIKV RNA levels. Data represent mean ±SEM of one representative experiment for A, three independent experiments for B and two independent experiments for C and D. (*p<0.05, **p<0.005, ***p<0.001). .

DOI: https://doi.org/10.7554/eLife.39023.013

The following source data is available for figure 4:

**Source data 1.**

DOI: https://doi.org/10.7554/eLife.39023.014

*KDM5C, TSC2, FXR2, TLN1* and *BRD4*. Protein levels of FXR2, TLN1 and BRD4 increased after transfection of siRNA targeting FMRP (*Figure 6—figure supplement 1*) suggesting that the mRNAs encoding these proteins are genuine FMRP targets in HeLa cells. The remaining proteins were either undetectable (ARC) or were unaffected (RAC1, Rho A, PNPLA6, KDM5C, TSC2, SOD1) which suggests that FMRP may have cell-type specific effects (*Ascano et al., 2012*; *Darnell and Klann, 2013*).

To probe for antagonistic effects of ZIKV sfRNA on FMRP activity, the expression of FMRP targets was evaluated in the context of infection. First, we analyzed expression levels of FXR2 by WB in HeLa cells infected with WT ZIKV and Δ10 ZIKV. In order to achieve similar infection rates for the two viruses, cells were infected at MOI of 3 for WT ZIKV and MOI of 4.5 for the mutant virus. Interestingly, these experiments revealed a 2.7-fold-increase of FXR2 protein levels in cells infected with WT ZIKV (*Figure 6A and B*). Although Δ10 ZIKV infection also increased the mean level of FXR2, this effect was not statistically significant (p=0.277) compared to uninfected cells. In parallel, we measured the expression of FXR2 by flow cytometry in ZIKV-infected cells using a double staining protocol. *Figure 6C* shows the distributions of the cells based on infection with either WT ZIKV or Δ10 ZIKV, indicating similar levels of infection for both viruses (~86% for WT and ~90% for Δ10 ZIKV). FXR2 mean fluorescence intensity (MFI) was enhanced by WT virus and, to a lesser extent, by the Δ10 ZIKV mutant. Quantitatively, there was a two-fold increase of FXR2 MFI in WT ZIKV infected cells and a 1.5-fold increase for Δ10 ZIKV compared to uninfected cells (*Figure 6D*). We also conducted IF microscopy analysis and observed FXR2 accumulation specifically in ZIKV-infected cells (*Figure 6E*). Interestingly, infection of cells by Δ10 ZIKV caused FXR2 to redistribute into discrete cytoplasmic puncta which was also evident at reduced frequency in WT ZIKV-infected cells. FXR2 has been previously reported to concentrate in stress granules (*Gonçalves et al., 2011*), suggesting that infection with the attenuated virus may induce formation of stress granules more efficiently than WT ZIKV. In addition to FXR2, we also observed elevated expression of TLN1 by WB in cells infected with WT ZIKV compared with uninfected and Δ10 ZIKV infected cells (*Figure 6—figure supplement 2A and B*).

We further assayed the expression of BRD4 in ZIKV-infected cells. Although the anti-BRD4 antibody used recognizes the largest known BRD4 isoform, we observed a second and faster migrating band of ~175 kDa that appeared only in ZIKV-infected cells (*Figure 6—figure supplement 2C*). Since there are no annotated isoforms of this predicted mass, the data suggest that BRD4 is cleaved during ZIKV infection. We analyzed the sum of signals for both 204 kDa and 175 kDa BRD4 species and observed a 1.6-fold increase of BRD4 expression only in WT ZIKV infected cells (*Figure 6—figure supplement 2D*).

## ZIKV sfRNA interferes with FMRP-mediated translational repression in uninfected cells

To further test the role of sfRNA we asked whether or not this noncoding RNA could modulate expression of FMRP targets in the absence of infection. Electroporation was used to transfect full-length, in vitro synthesized sfRNA or a deletion mutant lacking xrRNA1 and xrRNA2 (DBSLIII). We electroporated RNAs with either 5′ triphosphate or monophosphate into HeLa cells and evaluated RNA integrity and abundance side-by-side with sfRNA from infected cells. We detected three distinct sfRNA species in infected cells that have been previously reported for ZIKV: sfRNA1, sfRNA2 and sfRNA3 (*Figure 7A and B*) (*Filomatori et al., 2017*). Electroporated RNAs were partially degraded although some intact sfRNA remained even at 48 hr post-electroporation. In parallel, we

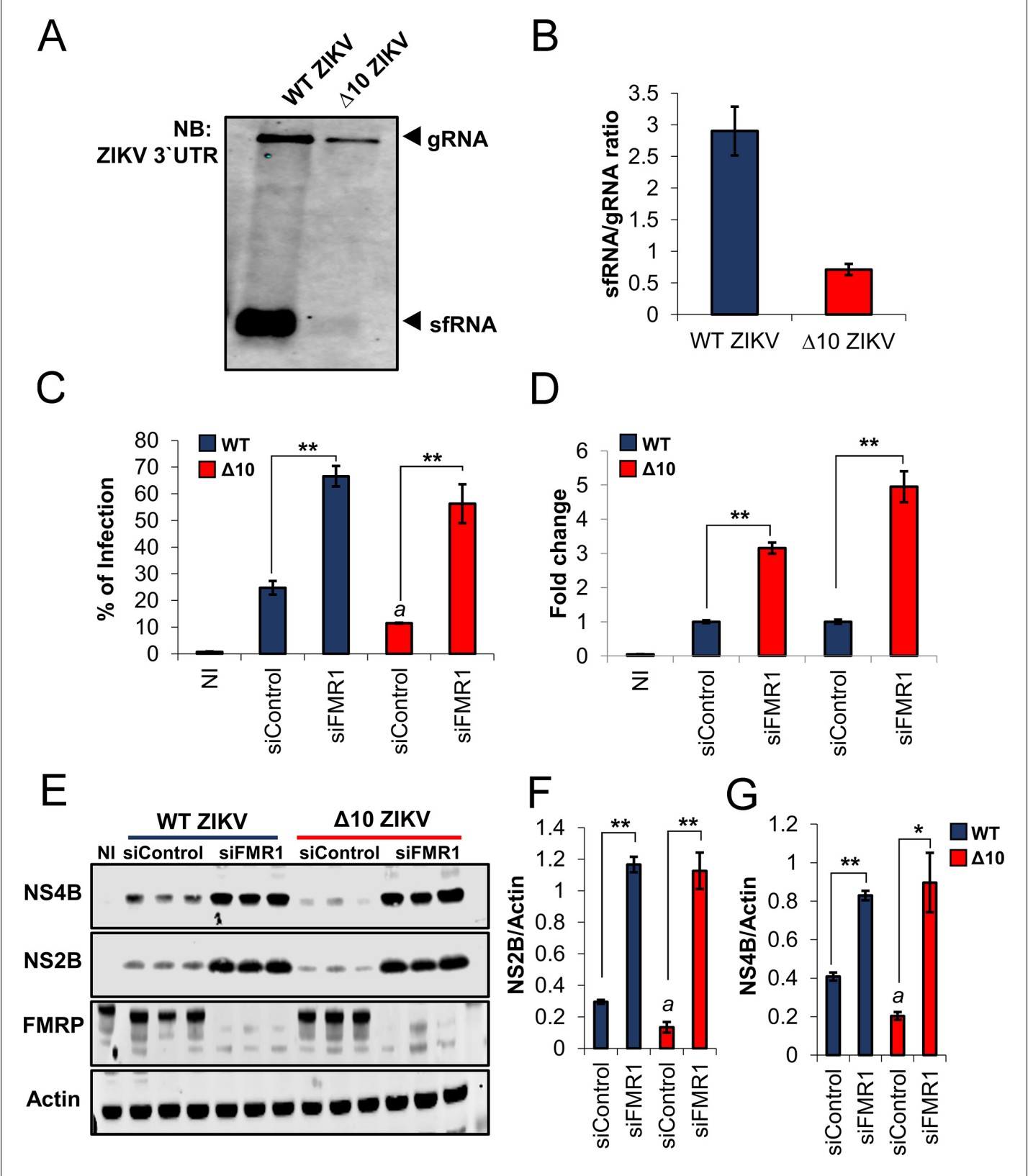

**Figure 5.** ZIKV sfRNA antagonizes FMRP's antiviral function. (**A**) HeLa cells were infected at MOI of 3 with WT ZIKV or Δ10 ZIKV. 24 hr post-infection, cell-associated RNA was harvested and levels of gRNA and sfRNA were analyzed by NB. (**B**) sfRNA/gRNA ratios for WT and Δ10 ZIKV were calculated by densitometry analysis of two independent experiments. (**C–G**) HeLa cells were transfected with control siRNA (siControl) and pooled FMR1 siRNAs

*Figure 5 continued on next page*

*Figure 5 continued*

for 48 hr. Cells were then infected with WT and Δ10 ZIKV at MOI of 0.2 and 24 hr later collected and analyzed for infection rate by flow cytometry (**C** and **D**). Representative scatter plots are shown in ***Figure 5—figure supplement 1***. In parallel, viral NS4B and NS2B proteins were analyzed by WB along with FMRP and Actin loading control (**E**). The quantitative expression levels of NS2B (**F**) and NS4B (**G**) are shown. Data represents the mean ±SEM of one representative experiment for C, four independent experiments for D and two independent experiments for F and G. (*p<0.05, **p<0.005, ***p<0.001. a = *p* < 0.01 compared to siControl cells infected with WT ZIKV). .

DOI: https://doi.org/10.7554/eLife.39023.015

The following source data and figure supplement are available for figure 5:

**Source data 1.**
DOI: https://doi.org/10.7554/eLife.39023.017
**Figure supplement 1.** Scatter plots of WT and Δ10 ZIKV infected cells.
DOI: https://doi.org/10.7554/eLife.39023.016

analyzed the expression of the FMRP targets FXR2 and BRD4 by flow cytometry (***Figure 7B*** to F). Compared to the DBSLIII RNA, electroporation of intact sfRNA resulted in modest, but highly significant, increases in both FXR2 and BRD4 levels. These results suggest expression of ZIKV sfRNA is sufficient to interfere with FMRP. We next asked whether ZIKV infection alters the expression of FMRP targets in mice (***Figure 8***). Male A129 mice were infected with WT (n = 4) or Δ10 ZIKV (n = 4) using 1 $\times$ $10^5$ FFU. At 6 days post-infection, protein expression levels of FXR2 and PNPLA6 [a validated FMRP target (***Ascano et al., 2012***)] were measured in testes. In WT ZIKV-infected mice, we observed statistically significant increases in expression of both FXR2 (1.6-fold) and PNPLA6 (2.5-fold) compared to uninfected mice (***Figure 8A,C and D***). For Δ10 ZIKV-infected mice, expression levels of FXR2 and PNPLA6 were elevated to lesser extents, but these effects were not statistically significant. Measurements of viral genomes indicated a higher burden of infection for the WT virus than Δ10 ZIKV (***Figure 8B***) which may contribute to the differential effects on FXR2 and PNPLA6 levels. Nevertheless, these data taken together with observations made in cultured cells (***Figures 6*** and ***7*** and ***Figure 6—figure supplement 2***), implicate the ZIKV sfRNA as an antagonist of FMRP function.

## Discussion

Here, we report functional interactions between ZIKV and FMRP, an important regulatory factor in neurodevelopment (***Harlow et al., 2010***; ***Hoeft et al., 2010***). FMRP interacted with ZIKV RNA, with particular affinity for the sfRNA, in both infected cultured cells and mouse testes. FMRP interaction with the viral genomic RNA presumably limits infection through inhibiting early synthesis of viral proteins. Importantly, accumulation of ZIKV sfRNA suppressed the anti-viral activity of FMRP and, consequently, resulted in de-repression of endogenous FMRP target mRNAs. Together, our observations have important implications for ZIKV infection and pathogenesis.

Multiple studies have been performed to characterize the RNA-binding specificity of FMRP (***Ascano et al., 2012***; ***Darnell et al., 2001***; ***Darnell et al., 2011***; ***Maurin et al., 2018***; ***Ray et al., 2013***). Bioinformatic analysis of FMRP CLIP-seq datasets identified WGGA (W = T/A) as the top sequence motif (***Anderson et al., 2016***) which is found at six sites in the ZIKV 3′ UTR, each of them located in the dumbbell regions. Our mapping experiments, however, showed that the dumbbells are dispensable for FMRP binding to ZIKV 3′UTR, suggesting that these WGGA motifs do not mediate interaction with FMRP. Interestingly, using in vitro RNA selection Darnell et al. identified a highly structured target for the FMRP KH2 domain containing a pseudoknot (***Darnell et al., 2005***). To date, no FMRP target mRNA has been identified that contains this motif but it is notable that the ZIKV xrRNA1, which is required for FMRP binding in vitro, folds into a complex structure containing several tertiary interactions (***Akiyama et al., 2016***). We speculate that the artificial RNA identified by Darnell and colleagues shares similar three-dimensional structural features with the sfRNA which allow it to bind FMRP with high affinity.

FMRP inhibited infection of multiple ZIKV strains but did not restrict the related flavivirus, DENV, even though we previously identified FMRP as a DENV RNA-binding protein by RNA affinity chromatography (***Ward et al., 2011***), suggesting that FMRP acts as an intrinsic restriction factor for ZIKV. It is not clear why DENV infection is immune to FMRP. It is possible that DENV eludes physical interaction with FMRP in infected cells or that ribosomes translating DENV RNA do not effectively recruit

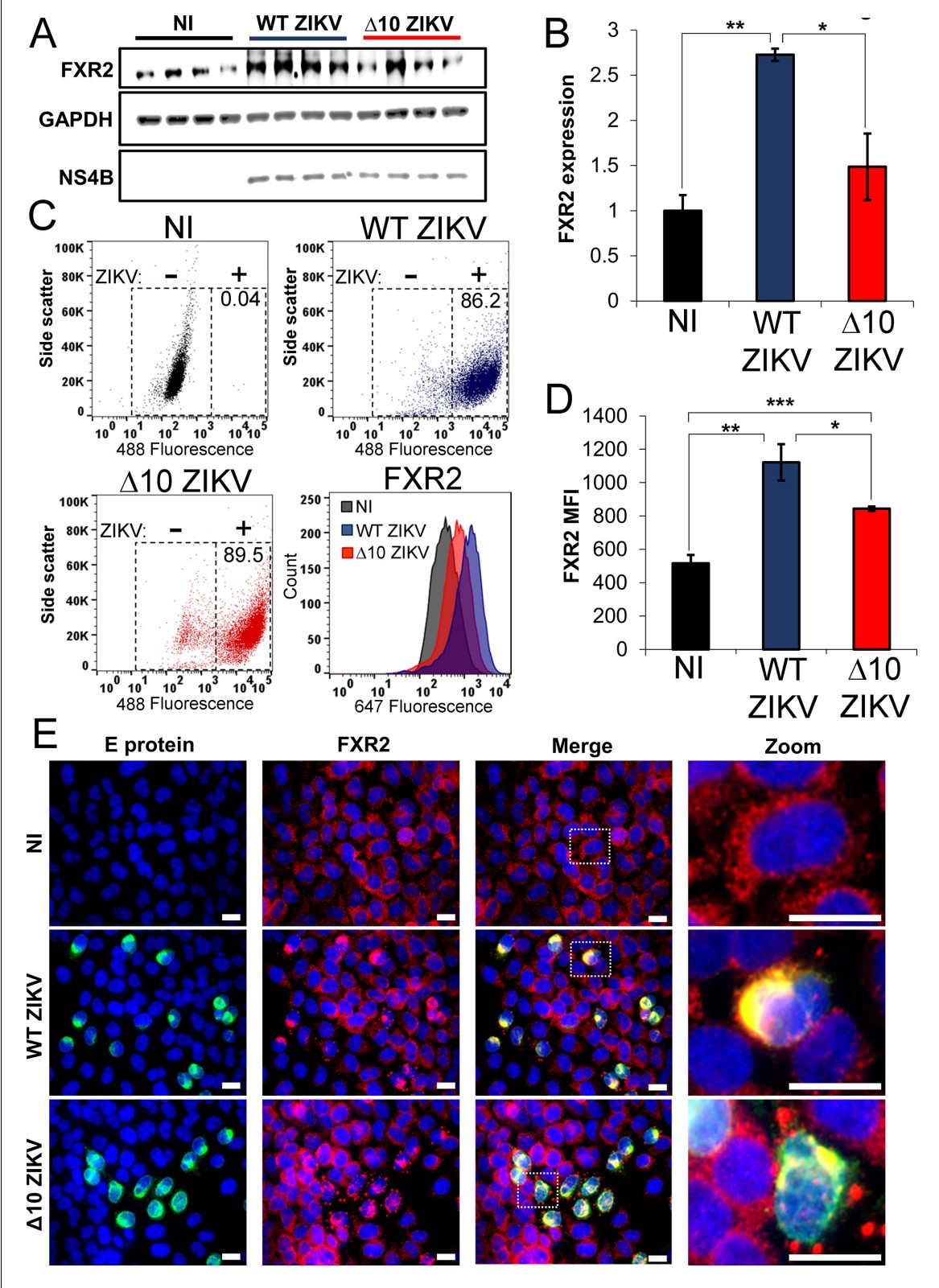

**Figure 6.** ZIKV infection increases expression of FXR2. HeLa cells were infected with WT (MOI of 3) or Δ10 ZIKV (MOI of 4.5) or left non-infected (NI). At 24 hr post-infection, cells were harvested and analyzed for FXR2 expression. (**A**) WB showing expression of FXR2, GAPDH and ZIKV NS4B. (**B**) Normalized expression levels of FXR2 from panel A are shown. (**C**) Flow cytometry for FXR2 and ZIKV E protein. E protein was stained using the 4G2 antibody and Alexa fluor 488-labeled secondary antibody. FXR2 protein was stained with anti-FXR2 and Alexa fluor 647-labeled secondary antibody.

*Figure 6 continued on next page*

*Figure 6 continued*

Scatter plots indicate the distribution of ZIKV positive cells in the NI cells (black, top left panel), WT ZIKV (blue, top right panel) and Δ10 ZIKV (red, bottom left panel) infected cells. Histograms (bottom right panel) indicate FXR2 expression as Alexa fluor 647 mean fluorescence intensity (MFI). (D) Graph shows the FXR2 MFI. Data is presented as the mean ±SEM of one of three independent experiments (*p<0.05, **p<0.005, ***p<0.001). (E) Representative immunofluorescence images from two independent experiments indicate the accumulation of FXR2 (red) in cells infected with WT and Δ10 ZIKV (green). Nuclei were counterstained with Hoechst 33342. White scale bar = 20 μm.
DOI: https://doi.org/10.7554/eLife.39023.018

The following source data and figure supplements are available for figure 6:

**Source data 1.**
DOI: https://doi.org/10.7554/eLife.39023.021
**Figure supplement 1.** FMRP knockdown increases protein levels of known FMRP targets.
DOI: https://doi.org/10.7554/eLife.39023.019
**Figure supplement 2.** Differential expression of TLN1 and BRD4 during infection with WT ZIKV and Δ10 ZIKV.
DOI: https://doi.org/10.7554/eLife.39023.020

FMRP, which is thought to be a prerequisite for translational repression (*Chen et al., 2014*; *Blackwell et al., 2010*). Notably, FMRP was recently described as a proviral host factor for influenza A virus that promotes assembly of viral RNPs, and FXR proteins were shown to redundantly promote replication of specific alphaviruses (*Zhou et al., 2014*; *Kim et al., 2016*). Hirano et al. showed interaction of FMRP with the genomic RNA of Tick-Borne Encephalitis Virus (TBEV), accumulation of FMRP at sites of local TBEV replication, and that FMRP depletion reduces TBEV infection (*Hirano et al., 2017*; *Muto et al., 2018*). Thus, FMRP can play a positive, negative or no role in infection, depending on viral species.

Mechanistically, we determined that FMRP inhibits ZIKV translation. FMRP is widely expressed and has been characterized to repress translation of specific neuronal mRNAs but the precise mechanism is unknown. Chen et al. demonstrated that drosophila FMRP (dFMRP) can bind directly to the ribosome in the absence of mRNA (*Chen et al., 2014*). The authors proposed that FMRP docks on the 80S ribosome using KH1/2 domains and simultaneously binds to mRNA via its RGG motif. The binding location of dFMRP on the ribosome suggests that it occludes recruitment of elongation factors and tRNA, leading the ribosome reversibly stall as observed by Darnell et al. (*Darnell et al., 2011*). Based on this proposed mechanism, we speculate that FMRP binds the ZIKV genome within the viral open reading frame and causes elongating ribosomes to stall, leading to a deficiency in viral protein synthesis. Alternatively, it is possible that FMRP interaction with the 3′ UTR of the viral genome could lead to impaired translation. Finally, changes in the expression of FMRP targets may indirectly contribute to enhanced ZIKV infection upon FMRP knockdown. For example, the increase in FXR2 subsequent to FMRP depletion could positively impact ZIKV infection (*Figure 3—figure supplement 2*).

We found that the recently developed ZIKV vaccine strain, Δ10 ZIKV, is compromised for sfRNA production and is replication-deficient in HeLa cells (*Shan et al., 2017a*; *Shan et al., 2017b*). Interestingly, infection with Δ10 ZIKV, but not WT ZIKV, caused relocalization of FXR2 to cytoplasmic granular structures reminiscent of stress granules (SG), suggesting that sfRNA may prevent SG formation. This is consistent with previous reports that show very low SG formation in flavivirus infected cells (*Bidet et al., 2014*; *Emara and Brinton, 2007*; *Ruggieri et al., 2012*). Importantly, Δ10 ZIKV infection was disproportionately increased by FMRP depletion compared to WT ZIKV, suggesting that one function of ZIKV sfRNA is to antagonize FMRP. In support of this, our RNA-IP experiments showed that FMRP preferentially co-IPs sfRNA compared to viral genomes. This may be partly explained by stoichiometry of sfRNA to genomes which reaches nearly 4 to 1 at the height of infection, depending on ZIKV strain. It is also possible that sfRNA structure differs from the corresponding region present in the viral genome and this dictates preferential FMRP interaction. We speculate that ZIKV sfRNA works as a 'sink' to saturate FMRP and prevent its repressive interactions with both viral RNA and cellular target mRNAs. This is reminiscent of recently described roles for the DENV sfRNA as a molecular sink for proteins involved in interferon responses (G3BP1, G3BP2, CAPRIN1, TRIM25) (*Bidet et al., 2014*; *Manokaran et al., 2015*).

Besides interferon responses, there are intrinsic restriction factors that recognize viral components and block replication (*Yan and Chen, 2012*). For flaviviruses, a few intrinsic anti-viral factors

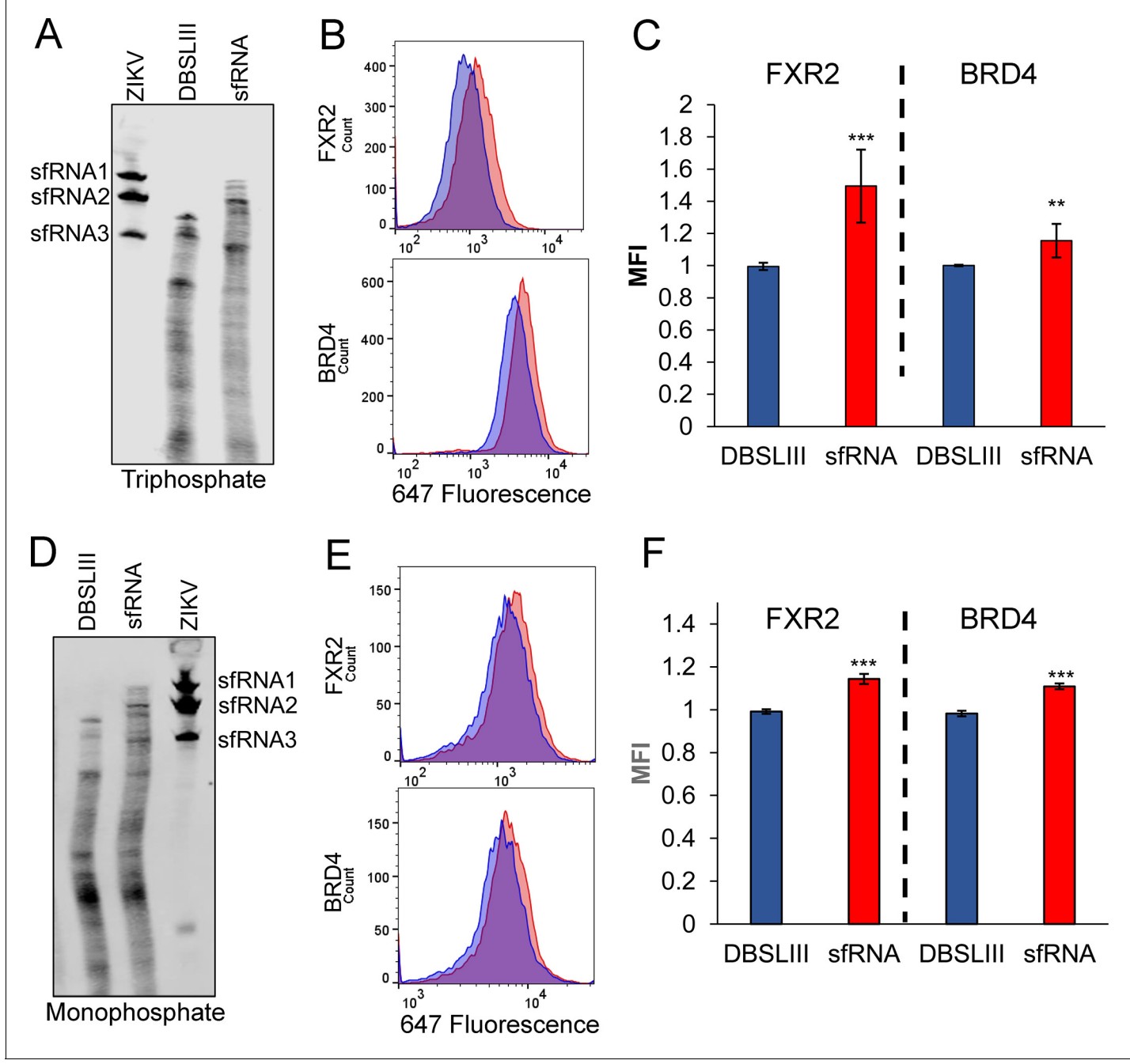

**Figure 7.** ZIKV sfRNA increases expression of FXR2 and BRD4 in the absence of infection. Electroporated 5' triphosphate (A) or monophosphate (D) RNAs were analyzed by northern blot, 48 hr after electroporation. Total RNA from cells infected with ZIKV-Puerto Rico was analyzed to visualize the presence of sfRNA1, sfRNA2 and sfRNA3. The expression levels of FXR2 and BRD4 were evaluated by flow cytometry in cells electroporated with 5' triphosphate (B,C) or monophosphate (E,F) RNAs. Mean fluorescence intensities (MFI) in presence of the mutant control DBSLIII (blue) and sfRNA (red) are shown. Plots in C and F represent the mean ±SD of four and two independent experiments, respectively. (*p<0.05, **p<0.005, ***p<0.001). MFI intensities were normalized to the mutant control. .

DOI: https://doi.org/10.7554/eLife.39023.022

The following source data is available for figure 7:

**Source data 1.**

DOI: https://doi.org/10.7554/eLife.39023.023

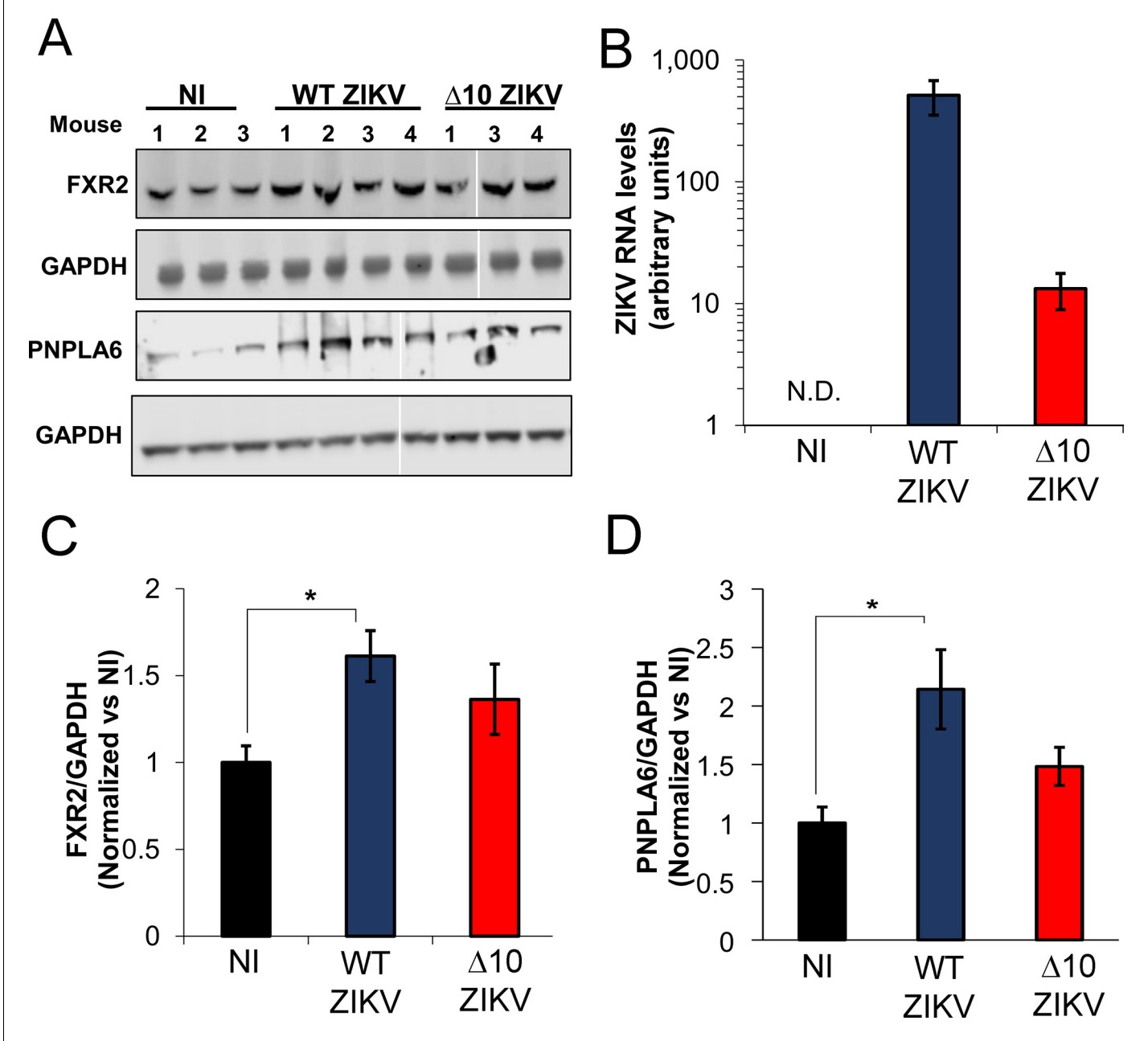

**Figure 8.** ZIKV infection increases the expression of FMRP targets in infected mouse testes. A129 mice were non-infected (NI; N = 3) or infected with WT (N = 4) or Δ10 ZIKV (N = 4) with $1 \times 10^5$ FFU. At 6 days post-infection, mice were euthanized, and testes were removed for lysis. (**A**) Protein expression levels of FXR2 and PNPLA6 were analyzed by WB. No FXR2 was detected in mouse #2 infected with Δ10 ZIKV and we therefore eliminated this sample from the analysis. (**B**) A fraction of testes lysates was processed for RNA isolation and levels of viral genomes were measured by RT-qPCR using a pair of primers that amplify a region of the ZIKV ORF. GAPDH was used as loading control for WB assays and calibrator gene for RT-qPCR. Relative protein expression levels of FXR2 (**C**) and PNPLA6 (**D**) were measured by densitometry analysis, adjusted to loading control and normalized to non-infected mouse (NI). Error bars represent the mean ±SEM of three mice for non-infected condition (NI), four mice for WT ZIKV and three mice for Δ10 ZIKV. *$p<0.05$. N.D, non-detected. .

DOI: https://doi.org/10.7554/eLife.39023.024

The following source data is available for figure 8:

**Source data 1.**

DOI: https://doi.org/10.7554/eLife.39023.025

have been described: YTHDF1-3 proteins that bind methylated RNA and regulate stability, were reported to interact with ZIKV RNA and inhibit infection (*Lichinchi et al., 2016*); YB-1 and QKI bind to the DENV 3′ UTR and repress DENV RNA translation (*Liao et al., 2018*; *Paranjape and Harris, 2007*); similarly, FBP1 blocks translation of JEV through interacting with untranslated regions of JEV RNA (*Chien et al., 2011*). The association of these antiviral factors with the viral genome and their functional consequences have only been observed in cell culture and it remains to be seen whether or not these will be validated in vivo. Furthermore, there are no reports that the intrinsic anti-viral function of these proteins is counteracted by flaviviral factors. In this work we describe that FMRP is an intrinsic, direct-acting ZIKV restriction factor that interacts with ZIKV RNA and is counteracted by the sfRNA as it accumulates during infection.

We provide evidence for de-repression of endogenous FMRP target mRNAs in the context of ZIKV infection of cultured cells and mouse testes. Specifically, we observed that the FMRP targets, *FXR2*, *TLN1*, *BRD4*, and *PNPLA6* were elevated at the protein level as a consequence of ZIKV infection. Moreover, infection with Δ10 ZIKV, which produces less sfRNA than WT virus, resulted in weaker or no effects on FMRP targets compared with WT ZIKV. We further observed that introduction of synthetic sfRNA into cells leads to upregulation of two FMRP targets: FXR2 and BRD4. These results provide functional evidence indicating that the ZIKV sfRNA interferes with the activity of FMRP, although sfRNA-independent mechanism(s) for modulation of FMRP targets by ZIKV may also be at play.

Our observations have implications for ZIKV pathogenesis in tissues with high expression of FMRP: brain, placenta and testes (*Hinds et al., 1993*). Given the well-established role for FMRP in promoting neurodevelopment, it is tempting to speculate that certain aspects of ZIKV neuropathogenesis may be explained by sfRNA-mediated FMRP inhibition, leading to inappropriate expression of FMRP target mRNAs. Development of congenital Zika syndrome is likely multifactorial and symptoms such as microcephaly, which is only one disease manifestation (*Aliota et al., 2017*), are unlikely to be the consequence of FMRP inhibition. Nevertheless, our findings warrant further research into how ZIKV interactions with FMRP might contribute to disease outcome.

## Materials and methods

### Key resources table

| Reagent type (species) or resource | Designation | Source or reference | Identifiers | Additional information |
|---|---|---|---|---|
| Genetic reagent (*Mus musculus*) | A129 IFNAR Knock out mouse | UTMB, Galveston, TX | MGI:5652612 | *Rossi et al. (2016)* PMID 27022155 |
| Antibody | 4G2 antibody; anti-Flavivirus envelope (Mouse monoclonal) | D1-4G2-4-15 Hybridome cell line | RRID for Hybridome cell line: CVCL_J890 | *Henchal et al. (1982)* PMID 6285749 |
| Antibody | Anti-FMRP (Rabbit polyclonal) | Abcam | Abcam Cat# ab17722, RRID:AB_2278530 | (1:1000) |
| Antibody | Anti-FXR1 (Rabbit monoclonal) | Cell Signaling Technology | Cat#12295S | (1:1000) |
| Antibody | Anti-FXR2 (Rabbit monoclonal) | Cell Signaling Technology | Cell Signaling Technology Cat# 7098S, RRID:AB_10891808 | (1:1000) |
| Antibody | Anti-G3BP1 (Rabbit polyclonal) | Bethyl | Bethyl Cat# A302-033A, RRID:AB_1576539 | (1:1000) |
| Antibody | Rabbit anti-PTB (Rabbit polyclonal) | Homemade | Wagner and Garcia-Blanco, 2002. PMID 12419237 | (1:4000) |
| Antibody | Anti-ZIKV NS4B (Rabbit polyclonal) | Genetex | GTX133321 | (1:1000) |

*Continued on next page*

*Continued*

| Reagent type (species) or resource | Designation | Source or reference | Identifiers | Additional information |
|---|---|---|---|---|
| Antibody | Anti-ZIKV NS2B (polyclonal) | Genetex | GeneTex Cat# GTX133308, RRID:AB_2715494 | (1:1000) |
| Antibody | Anti-BRD4 (Rabbit monoclonal) | Cell Signaling Technology | Cell Signaling Technology Cat# 13440, RRID:AB_2687578 | (1:1000) |
| Antibody | Anti-GAPDH (Rabbit polyclonal) | Abcam | Abcam Cat# ab9485, RRID:AB_307275 | (1:4000) |
| Antibody | Anti-TLN1 (Mouse monoclonal) | Santa Cruz Biotechnology | Santa Cruz Biotechnology Cat# sc-365875, RRID:AB_10842054 | (1:300) |
| Antibody | Anti-PNPLA6 (Mouse monoclonal) | Santa Cruz Biotechnology | Santa Cruz Biotechnology Cat# sc-271049, RRID:AB_10610321 | (1:1000) |
| Antibody | Anti-DDX6 (Rabbit polyclonal) | Cell Signaling Technology | Cell Signaling Technology Cat# 9407S, RRID:AB_10556959 | (1:1000) |
| Sequence-based reagent | siFMR1_10 | Qiagen | SI04916436 | TCCAAGGAACTTAGTAGGCAA |
| Sequence-based reagent | siFMR1_4 | Qiagen | SI00031626 | TCCGTAATTCTTATTCCATAT |
| Sequence-based reagent | siFMR1_3 | Qiagen | SI00031619 | CTCGTGAATGGAGTACCCTAA |
| Sequence-based reagent | siFMR1_2 | Qiagen | SI00031612 | CTGTCAAACATTAGTACTTTA |
| Sequence-based reagent | siFXR1_5 | Qiagen | SI03040429 | ACCGTCGTAGGCGGTCTCGTA |
| Sequence-based reagent | siFXR1_3 | Qiagen | SI00072247 | GTGGTTCGAGTGAGAATTGAA |
| Sequence-based reagent | siFXR1_2 | Qiagen | SI00072240 | CAGCTAAGAATGGAACGCCTA |
| Sequence-based reagent | siFXR1_1 | Qiagen | SI00072233 | ATGGAATGACTGAATCTGATA |
| Sequence-based reagent | siFXR2_8 | Qiagen | SI04347833 | TGGGTGATATGCATTTCCGAA |
| Sequence-based reagent | siFXR2_7 | Qiagen | SI04332454 | CTGGAACGCACTAAACCCTCA |
| Sequence-based reagent | siFXR2_6 | Qiagen | SI04316445 | AAACGTCCATAAAGAGTTCAA |
| Sequence-based reagent | siFXR2_5 | Qiagen | SI04284763 | TGGAGCGACTTCGGCCAGTTA |
| Commercial assay or kit | Northern Max kit | Thermo Fisher Scientific | Cat#:AM1940 | |
| Commercial assay or kit | HiScribe T7 High Yield RNA Synthesis Kit | New England Biolabs | Cat#: E2040S | |
| Chemical compound, drug | NITD008 | Other | CAS number: 1044589-82-3; Pubchem ID: 44633776 | *Yin et al. (2009)* PMID 19918064 |

## Cell culture, viral stocks and infection

JEG3 (human choriocarcinoma), HeLa (human cervix adenocarcinoma) and Vero cells were maintained in DMEM (Thermo Fisher Scientific) supplemented with 10% fetal bovine serum (FBS) and $1 \times$ penicillin/streptomycin (pen-strep) (Thermo Fisher Scientific) in a humidified incubator at 37°C with 5% $CO_2$. C6/36 cells (Aedes albopictus) were grown in RPMI medium (Thermo Fisher Scientific) supplemented with 10% FBS (GenDEPOT) and $1 \times$ pen strep and incubated at 28°C with 5% $CO_2$. Cells were tested for mycoplasma contamination every three months. HeLa and JEG3 cells were authenticated by STR analysis at the UTMB Molecular Genomics Core. Dakar 41525 (ZIKV-Dakar), Cambodia FSS13025 (ZIKV-Cambodia) and PRVABC59 (ZIKV-Puerto Rico) strains were kindly provided by Scott Weaver (UTMB) and Nikos Vasilakis (UTMB) and propagated in C6/36 cells. DENV-2-NGC (New Guinea C strain) was propagated as previously described (*Sessions et al., 2009*). Nano-luciferase reporter ZIKV was engineered following the same strategy reported in Xie et al. (*Xie et al., 2016*). Δ10 ZIKV and its WT version (ZIKV-Cambodia) were derived from infectious clones and propagated in Vero cells as described in (*Shan et al., 2017b*). Viral infections were performed in DMEM supplemented with 1% FBS. After 1 hr of incubation, media was replaced with complete DMEM media. MOIs and post-infection times are specified in the figures.

## Virus titration

ZIKV stocks and supernatants from *Figure 3* were analyzed by plaque assay in Vero cells (*Agbulos et al., 2016*). Titers of DENV-2 stock, Δ10 ZIKV and WT ZIKV from infectious clones were performed in Vero cells using foci-forming assay as previously described (*Shan et al., 2017b*; *Sessions et al., 2009*).

## Antibodies

The following antibodies were used for IP, western blotting and/or immunofluorescence analysis: anti-envelope protein 4G2 *Henchal et al., 1982*, rabbit IgG (2729S, Cell Signaling Technologies), anti-FMRP (ab17722, ABCAM, Cambridge, UK), anti-FXR1 (12295S, Cell Signaling Technologies), anti-FXR2 (7098S, Cell Signaling Technologies), anti-G3BP1 (A302-033A, Bethyl Laboratories), rabbit anti-DDX6 (9407S, Cell Signaling technologies), rabbit anti-PTB (homemade), rabbit anti-ZIKV NS4B (GTX133321, Genetex), anti-ZIKV NS2B (GTX133308, Genetex), anti-BRD4 (13440, Cell Signaling Technologies), anti-GAPDH (ab9485, ABCAM), anti-TLN1 (SC-365875, Santa Cruz Biotechnology), anti-PNPLA6 (SC-271049, Santa Cruz Biotechnology).

## RNA affinity chromatography and mass spectrometry

Synthetic DNA fragments corresponding to the complete 3' UTRs of African (strain MR766, accession number KX377335.1) and pandemic Asian-lineage ZIKV (PRVABC59 strain, KX377337.1) were obtained from IDT (Integrated DNA Technologies) and cloned into pcDNA3.1(+) plasmid containing tobramycin aptamer (*Liao et al., 2018*). As a control RNA we used a sequence corresponding to the DENV NS2A sequence as previously reported (*Manokaran et al., 2015*). PCR was used to generate DNA templates for T7 in vitro transcription using the MEGAscript T7 kit (Thermo Fisher Scientific). RNA affinity chromatography using JEG3 cell lysate was performed following the published protocol (*Ward et al., 2014*). Eluted RBPs were identified by label-free mass spectrometry at the UTMB Mass Spectrometry Core. Samples were dissolved in 30 µl denaturation buffer (4% SDS and 100 mM DTT in 0.1M TEAB pH 8.5), heated at 65°C for 15 min, and loaded onto 30 kDa spin filters (Merck Millipore). The buffer was exchanged three times with UA solution (8 M UREA in 0.1 M TEAB pH 8.5) by centrifugation at 14,000 g. After removal of SDS, cysteine alkylation was accomplished through the addition of alkylation buffer (50 mM IAA, 8 M UREA in 0.1 M Tris-HCl pH 8.5) for 1 hr at room temperature in the dark. UA buffer was exchanged with TEAB buffer (40 mM TEAB pH 8.5). The proteins were digested with trypsin (enzyme-to-substrate ratio [w/w] of 1:100) and 5% ACN at 37°C overnight. Peptides were centrifuged through the size exclusion membrane and collected into a clean microcentrifuge tube, followed by a rinse with 80 µL of 0.2% formic acid. The combined peptide solution was then dried in a speed vac and resuspended in 2% acetonitrile, 0.1% formic acid, 97.9% water and placed in an autosampler vial for LC/MS analysis. Raw data files from the mass spectrometer were aligned by accurate mass and time in Progenesis QI for proteomics (version 2.0.5556.29015, Nonlinear Dynamics, a Waters Company). The top five spectra for each feature

were exported as *.mgf files and searched against a combined Uniprot-Human canonical database. Peptide identifications were imported into Progenesis QI, and all peptides with −10logP scores < 30 (Mascot or confidence score) were removed. Conflict resolution was performed in order to remove lower scoring peptides when multiple peptides were assigned to a single feature. Protein quantification was calculated from normalized peptide abundances using a summed abundance of unique peptides. Unique peptide abundance was calculated by the area of the corresponding peaks in the ion chromatograms. Protein abundance from RBPs interacting with PRVABC59 and MR766 ZIKV 3′ UTRs were compared with control RNA to calculate ratios of enrichment (PRVABC59 3′ UTR/NS2A and MR766 3′ UTR/NS2A). Eight proteins that were identified by $\geq$2 unique peptides and were found $\geq$2 times in both the ZIKV sequences relative to the control RNA were considered high-confidence ZIKV 3′ UTR binding proteins RBPs with ratio >2 were considered as enriched.

## Immunoprecipitation

$1.5 \times 10^6$ HeLa cells were plated in 10 cm dishes. 24 hr later, cells were infected with ZIKV-Dakar, ZIKV-Cambodia and ZIKV-Puerto Rico at an MOI of 3 for 48 hr. Cells were washed three times with cold PBS, scraped and pelleted in 3 mL of PBS (5 min, 1500 rpm at 4°C). Cell pellets were resuspended in an equal volume of RIPA buffer (Cell Signaling Technologies) with protease inhibitors (Roche). All steps of the IP were performed at 4°C. For each IP 100 μL of protein A/G PLUS-Agarose (Santa Cruz Biotechnology) was washed with 1 ml NT2 buffer (50 mM Tris-HCl pH 7.4, 150 mM NaCl, 1 mM $MgCl_2$ and 0.05% IGEPAL) and blocked for 30 min in NT2 buffer with 0.5 mg/mL BSA. Beads were washed three times with NT2 buffer and incubated with either 10 μg of rabbit anti-FMRP, anti-PTB or rabbit IgG in a total volume of 600 μL of NT2 buffer for 2 hr with head to tail rotation. After coupling antibodies, beads were washed three times with NT2 buffer. One mg of protein from clarified lysate was diluted in NT2 buffer in a total volume of 600 μL and added either to IgG or FMRP beads and incubated with rotation for 1 hr. After incubation, beads were pelleted, washed four times with NT2 buffer and resuspended in 115 μL of NT2. 15 μL of beads were used for WB and 100 μL for RNA isolation using Trizol. Equal volumes of RNA (5 μL) from IP reactions were processed for Northern blotting.

## Northern Blot

The NorthernMax kit (Thermo Fisher Scientific) was used following the manufacturer´s instructions. RNA was mixed with 3 volumes of formaldehyde loading dye, then incubated 15 min at 65°C and 2 min on ice. Electrophoresis was performed in denaturing 1% agarose gels at 95 V. After electrophoresis, the gel was incubated in alkaline buffer (0.01N NaOH, 3M NaCl) for 20 min and transferred to a Biodyne B nylon membrane (Thermo Fisher Scientific) by downward transfer. RNA was crosslinked to the membrane using the UV stratalinker 2400 auto-crosslink program. The membrane was pre-hybridized for 30 min with ULTRAhyb-oligo Buffer (Thermo Fisher Scientific) at 42°C and hybridized overnight at 42°C with a Biotin-labeled DNA probe to detect ZIKV 3′ UTR (10453–10739 nt). The DNA probe was generated by PCR using 60% of unmodified dTTP and 40% biotin-16-dUTP (Roche). Specific primers (forward: 5′ AGGAGAAGCTGGGAAACCAAGC 3′; reverse: 5′ GATAATACGAC TCACTATAGGAAACTCATGGAGTCTCTGGTC 3′) and apt-MR766 3′ UTR plasmid (template) were used to generate the biotin-labeled probe. For *Figure 7* northern blots, RNA was mixed with 2X TBE-Urea loading sample and resolved in 6% acrylamide TBE, 8M Urea gels and transferred to nylon membrane in 0.5% TBE for 3 hr using the XCell II Blot module (Thermo Fisher Scientific). The membrane was crosslinked and hybridized with a DNA probe to detect the ZIKV 3′ UTR DB and SLIII (10528–10807 nt). This probe was generated as above using specific primers (forward: 5′ GATAA TACGACTCACTATAGGGCCCCTCAGAGGACACTGAGTCAAAAAA 3′; reverse: 5′ AGAAACCA TGGATTTCCCCACACCGGCCGCCGCT 3′). After hybridization, the membranes were washed and incubated for 1 hr at room temperature with IRDYE 800CW streptavidin (LI-COR Biosciences) in Odyssey Blocking Buffer (LI-COR Biosciences) with 1% of SDS. After three washes with TBS containing 0.1% tween, the membrane was scanned using a LI-COR Odyssey instrument. Densitometry was performed using Image Studio Lite Ver 5.0 software.

## siRNA transfection

Protein knockdown was performed using individual siRNAs (Qiagen) or a pool of siRNAs. $3 \times 10^4$ HeLa cells were plated in a 24 well plate. The next day cells were transfected with AllStars Negative Control siRNA (Qiagen), individual siRNAs, or the pool using 30 nM of siRNA and 1.5 µL of RNAi-MAX (Thermo Fisher Scientific) diluted in 50 µL of Opti-MEM media (Thermo Fisher Scientific). After 4 hr, transfection media was replaced by fresh media and cells were incubated for 48 hr. HeLa cells were infected as noted in the figure legends.

## Western blotting

Cells were lysed in RIPA buffer (Cell Signaling Technologies) with 1X protease inhibitor. 10 µg of protein samples were resolved under denaturing conditions on 4–12% acrylamide gels (Thermo Fisher Scientific). ZIKV NS4B, and NS2B, BRD4, TLN1, PNPLA6, Actin and GAPDH were detected using primary antibodies mentioned above. Goat anti-mouse IgG and goat anti-rabbit IgG coupled with IRDye 800 or 680 (LI-COR Biosciences) were used as secondary antibodies. Blots were developed in the LI-COR Odissey luminescence system and protein expression was quantified by densitometry with the Image Studio Lite Ver 5.0 software.

## Flow cytometry

Cells were harvested and fixed with 1% formaldehyde, permeabilized and blocked for 20 min (1X PBS, 0.1% tween 20, 5% FBS), incubated for 1 hr at room temperature with mouse 4G2 antibody and then 1 hr with goat anti-mouse IgG Alexa fluor 647 (Thermo Fisher Scientific) to detect ZIKV envelope protein (E protein). E protein fluorescence was measured with the Guava easyCyte system (Millipore) using the red laser (642 nm) for *Figures 3* and *5* and *Figure 3—figure supplements 1–3*. Double staining was performed for *Figure 6* to detect E protein and cellular FXR2 protein. E protein was detected using mouse 4G2 antibody and goat anti-mouse IgG Alexa fluor 488. FXR2 protein was stained using rabbit anti-FXR2 and goat anti-rabbit IgG Alexa fluor 647. Sample analysis, scatter plots of infected cells, mean fluorescence intensities and histograms were performed in FlowJo V10 software.

## Immunofluorescence staining

To determine percentage of infection, $3 \times 10^4$ HeLa cells per well were plated in 24-well plates. 24 hr later, cells were transfected with control siRNA or FMR1 siRNA (pool). After 48 hr of knockdown, cells were infected with ZIKV-Dakar for 48 hr, fixed with 1% paraformaldehyde, permeabilized and blocked for 20 min (1 × PBS, 0.1% tween 20, 1% FBS), and incubated for 1 hr at room temperature with mouse 4G2 mouse (E protein) and 1 hr with Goat anti-mouse IgG Alexa fluor 647. Nuclei were counterstained with Hoechst 33342 (Sigma-Aldrich). Image acquisition and infection rates were calculated using a high-content imaging microscope (Opera Phenix, Perkin Elmer). To visualize the accumulation of FXR2 protein in infected cells, glass slides with non-infected or ZIKV infected cells (WT or Δ10 ZIKV) for 24 hr were fixed, permeabilized and blocked as described above. Viral protein was detected using 4G2 antibody and goat anti-mouse IgG Alexa fluor 488. FXR2 protein was stained using rabbit anti-FXR2 and goat anti-rabbit IgG Alexa fluor 647. Nuclei were counterstained with Hoechst 33342 (Sigma-Aldrich) and images were acquired using an Olympus fluorescence microscope.

## Evaluation of viral translation using NanoLuc ZIKV reporter

$5 \times 10^4$ HeLa cells were plated in 24-well plates. Next day, cells were transfected with control siRNA or FMR1_2 siRNA. After 46 hr of knockdown, cells were treated with 0.05% DMSO or 20 µM NITD008 for 2 hr before infection with the ZIKV reporter at an MOI of 3. NITD008 or DMSO were retained in media during the infection. After 1 hr of incubation, inoculum was retired and replaced with fresh media containing DMSO or NITD008. 2.5 hr later, cells were washed five times with 1 × PBS and lysed with *Renilla* lysis buffer (Promega). Additionally, cycloheximide (CHX) treatment (200 µM) was used as a background control in absence or presence of NITD008. For this condition, cells were pretreated with CHX (in absence or presence of NITD008) for 2 hr before ZIKV reporter infection and CHX was retained during infection until cell lysis. Luciferase assays were performed

using the High-Affinity NanoBit evaluation system (Promega) and the Enspire plate reader (Perkin Elmer).

## Viral RNA quantitation

Viral genome and GAPDH RNA were quantified by RT-qPCR. Cell-associated RNA from HeLa was extracted by the Trizol method (Thermo Fisher Scientific) and reverse transcribed using the Multi-Scribe Reverse transcriptase protocol (Thermo Fisher Scientific). qPCR was performed with SYBR mix on a StepOne plus instrument (Thermo Fisher Scientific). The ΔΔCT method was used to calculate relative expression levels of viral genome. Primers used for ZIKV ORF (nucleotides 4541 to 4631 of ZIKV-Cambodia: forward 5′ CTGTGGCATGAACCCAATAG 3′; reverse 5′ ATCCCATAGAGCACCAC TCC 3′). Primers used for human GAPDH (forward 5′ AGCCACATCGCTCAGACAC 3′; reverse 5′ GCCCAATACGACCAAATCC 3′). To evaluate viral genome in mice, RNA from testes lysates were obtained by diluting 15 µL of cell lysate in 85 µL of NT2 buffer and 300 µL of Trizol LS. RNA transcription and qPCR were performed as mentioned above using primers for ZIKV ORF and mouse GAPDH (forward 5′ AAGGTCATCCCAGAGCTGAA 3′; reverse 5′ AAGGTCATCCCAGAGCTGAA 3′).

## Evaluation of FMRP-viral genome interaction and FMRP targets in mice

All animal studies were done in accordance with IACUC protocols as per UTMB policy. A129 mice were obtained from colonies maintained under specific pathogen-free conditions. Male 8–9 week-old A129 mice were infected with $1 \times 10^5$ FFU of WT (n = 4) or Δ10 ZIKV (n = 4) mutant viruses through the intraperitoneal route. PBS was given to the mock-infected mice through the same route (n = 3). At 6 days post-infection mice were euthanized, and testes were removed immediately as previously described (Hansen et al., 2014). Testes were flash-frozen in dry ice and stored at −80°C. Tissue was homogenized and lysed with a tissue grinder (OmniTHQ) in 500 µL of polysome lysis buffer (10 mM HEPES pH 7.0, 100 mM KCl, 25 mM EDTA, 5 mM MgCl₂,1 mM DTT, 0.5% NP-40). RNasin 1:1000 dilution (Promega) and protease inhibitor (Roche) were freshly added to samples. Samples were rotated for 10 min at 4°C to induce lysis and then flash-frozen on dry ice. Samples were thawed and nuclei were pelleted at 3000 x g for 10 min. 10 µg of protein samples from tissue lysates were fractionated in a 4–12% acrylamide gel (Thermo Fisher Scientific), transferred to a nitrocellulose membrane and blotted for FXR2, PNPLA6 and GAPDH with antibodies mentioned above. For RIP, testis lysate obtained from a mouse infected with WT ZIKV (ZIKV-Cambodia) was pre-cleared by adding 50 µL of protein A/G PLUS-Agarose (Santa Cruz Biotechnology) and rotating for 30 min at 4°C. 50 µL of beads were blocked in 600 µL of NT2 buffer (50 mM Tris-HCl, pH 7.4, 150 mM NaCl, 1 mM MgCl₂, 1 mM DTT, 0.05% NP-40) with 0.5 mg/mL BSA (Thermo Fisher Scientific) and 0.5 mg/mL yeast tRNA (Sigma-Aldrich). After blocking, beads were washed and incubated overnight at 4°C with 10 µg of either FMRP or Rabbit IgG antibodies. Antibody-bound beads were washed four times with ice-cold NT2 buffer. For the IP, 2 mg of pre-cleared protein was added to the antibody-bound beads. Final volume was adjusted to 600 µL with NT2 buffer containing RNase and protease inhibitors and incubated for 1.5 hr at 4°C. After incubation, beads were washed four times with NT2 buffer and processed for WB and RNA isolation as in the RIP for HeLa cells. qPCR was performed using ZIKV ORF primers described above and specific primers that amplify ZIKV 3′ UTR (nucleotides 10623 to 10722 of ZIKV-Cambodia: forward 5′ CCTGAACTGGAGATCAGCTGTG 3′; reverse 5′ GGTCTTTCCCAGCGTCAATA 3′).

## RNA electroporation

5′ triphosphate RNAs were generated as described above using the HiScribe T7 High Yield RNA Synthesis Kit (New England Biolabs) and specific primers for sfRNA (forward: 5′ GATAATACGAC TCACTATAGGGT GTTGTCAGGCCTGCTAGTCAGCCACAGC 3′; reverse: 5′ AGAAACCATGGA TTTCCCCACACCGGCCGCCGCT 3′) and the DBSLIII mutant (forward: 5′ GATAATACGACTCACTA TAGGGCCCCTCAGAGGACACTGAGTCAAAAAA 3′). 5′ monophosphate RNAs were by generated by synthesis of capped RNAs followed by treatment with Tobacco Acid Pyrophosphatase (Thermo Fisher Scientific). 5′ tri- and monophosphate RNAs were treated with the Turbo DNAse I (Thermo Fisher Scientific). RNAs were purified using Trizol LS and Direct Zol RNA MiniPrep (Zymo Research) for RNA clean up. 36 pmol of DBSLIII or sfRNA were electroporated into $2 \times 10^6$ HeLa cells in 250 µL of Ingenio electroporation solution (Mirus) using the Gene Pulser X cell Electroporation System

(Bio-Rad Laboratories) following manufacturer´s instructions. 48 hr post electroporation, cells were analyzed for expression of FXR2 and BRD4 and transfection efficiency by NB as described above.

## Experimental design and statistical analysis

The independent experiments were performed with three biological replicates except for *Figure 2A and B*. For *Figure 2C* two infected mice, each with three technical replicates, were processed separately. Differences between treatments and control groups were evaluated using the SigmaPlot/Stat package 11. In all cases, parametric or nonparametric tests and the appropriate post-hoc test were applied. If data met with assumptions of normality (Shapiro-Wilk test) and equal variance test, a *t*-test (parametric) was conducted. Data that did not meet with either normality test or equal variance test were analyzed using Mann-Whitney U test.

## Acknowledgments

We thank Drs. William Russell, Alexander Shavkunov and Cheryl Lichti of the UTMB Mass Spectrometry Facility for the sample processing, analysis and discussion regarding mass spectrometry. This work was supported by NIH grants R01AI089526, R01AI101431 and UTMB start-up funds (MAG-B and SSB). P-YS lab was supported by a University of Texas STARs Award, an award from the Robert J. Kleberg, Jr. and Helen C. Kleberg Foundation, UTMB CTSA UL1TR-001439, and NIH grant R01AI127744. The funding agency had no role in study design, data collection and interpretation, or the decision to submit the work for publication.

## Additional information

### Funding

| Funder | Grant reference number | Author |
| --- | --- | --- |
| National Institute of Allergy and Infectious Diseases | R01AI089526 | Mariano A Garcia-Blanco |
| University of Texas Medical Branch | Start-up funds | Mariano A Garcia-Blanco Shelton Bradrick |
| National Institute of Allergy and Infectious Diseases | R01AI101431 | Mariano A Garcia-Blanco |
| National Institute of Allergy and Infectious Diseases | R01AI127744 | Pei-Yong Shi |
| University of Texas System | STARs award | Pei-Yong Shi |
| Robert J. Kleberg, Jr. and Helen C. Kleberg Foundation | Award | Pei-Yong Shi |
| University of Texas Medical Branch | CTSA UL1TR-001439 | Pei-Yong Shi |

The funders had no role in study design, data collection and interpretation, or the decision to submit the work for publication.

### Author contributions

Ruben Soto-Acosta, Conceptualization, Investigation, Visualization, Methodology, Writing—original draft, Writing—review and editing; Xuping Xie, Chao Shan, Coleman K Baker, Resources, Methodology, Writing—review and editing; Pei-Yong Shi, Resources, Funding acquisition, Methodology, Writing—review and editing; Shannan L Rossi, Resources, Investigation, Methodology, Writing—review and editing; Mariano A Garcia-Blanco, Conceptualization, Supervision, Funding acquisition, Writing—original draft, Project administration, Writing—review and editing; Shelton Bradrick, Conceptualization, Supervision, Funding acquisition, Visualization, Writing—original draft, Project administration, Writing—review and editing

## Author ORCIDs
Ruben Soto-Acosta (iD) http://orcid.org/0000-0003-0570-5483
Chao Shan (iD) http://orcid.org/0000-0002-3953-3782
Mariano A Garcia-Blanco (iD) https://orcid.org/0000-0001-8538-1997
Shelton Bradrick (iD) http://orcid.org/0000-0003-1566-9797

## Ethics

Animal experimentation: Animal studies were done in accordance with IACUC protocols as per UTMB policy (Protocol number: 0209068B). All efforts were made to minimize animal suffering.

## Decision letter and Author response

Decision letter https://doi.org/10.7554/eLife.39023.028
Author response https://doi.org/10.7554/eLife.39023.029

## Additional files

### Supplementary files
• Transparent reporting form
DOI: https://doi.org/10.7554/eLife.39023.026

### Data availability

All data generated or analysed during this study are included in the manuscript and supporting files. Source data have been provided for Table 1.

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
