## [Decision Letter]

Thank you for submitting your article "Fragile X mental retardation protein is Zika virus restriction factor that is antagonized by subgenomic flaviviral RNA" for consideration by *eLife*. Your article has been reviewed by three peer reviewers, and the evaluation has been overseen by a Reviewing Editor and Wenhui Li as the Senior Editor. The reviewers have opted to remain anonymous.

The reviewers have discussed the reviews with one another and the Reviewing Editor has drafted this decision to help you prepare a revised submission.

Summary:

The translational repressor protein FMRP has been previously implicated as a host factor that modulates viral infectivity, for example as a Dengue virus (DENV) RNA binding protein (RBP), an influenza A RNP assembly factor and a positive host factor for Tick-Borne Encephalitis Virus infection. Additionally, the FMRP-related FXR proteins enhance replication of some alphaviruses. In this manuscript, Soto-Acosta and coworkers add to this list by using RNA affinity chromatography and MS to identify a number of host proteins that bind to the Zika virus 3'UTR. Among these proteins FMRP, and related proteins FXR1 and FXR2, were selected for further studies. These proteins bind a specific RNA structure of the viral 3'UTR known as xrRNA1, which is present in the viral genome and in viral sfRNAs. FMRP, a cellular RNA-binding protein (RBP) that binds to the 3'UTR of Zika virus, inhibits virus production by interfering with viral translation. FMRP binds a specific 3' UTR region (xrRNA1) of ZIKV subgenomic RNA (sfRNA), which accumulates during ZIKV infection and inhibits FMRP inhibitory activity. Using siRNA-mediated FMRP depletion, they provide evidence that FMRP loss increases ZIKV translation/infection and also stimulates sfRNA-deficient ZIKV10-del. ZIKV also upregulates FMRP translational targets in both Hela cells and mouse testes.

Studies on ZIKV continue to be of biomedical concern due to ZIKV-associated congenital microcephaly and inflammatory peripheral neuropathy. In this context, the findings reported in this study are important since they add to our understanding by dissecting the interaction of FMRP/FXR proteins with sfRNA. Overall, the described experiments are well-controlled, the manuscript is well-written and the finding that FMRP acts as a restriction factor for ZIKV infection is novel and of high interest. The reviewers felt, however, that key experiments were missing to fully support the conclusions. Especially, the conclusion that sfRNA strongly antagonizes FMRP activity is less well supported because it is mainly based on an attenuated ZIKV strain with distinct replication kinetics.

Essential revisions:

1) Further testing of the specificity of the FMRP-sfRNA interaction is needed.

1a) The Δ10 mutation produces less sfRNAs but its attenuation can be due to any number of reasons. For example, the 3'UTR is crucial for recruiting pro-viral RBPs that aid in translation and replication; the Δ10 mutation can disrupt this independently of sfRNAs. Thus, the observation that the attenuated virus is rescued to a higher extent upon FMRP depletion could be due to other non sfRNA related mechanism. More direct evidence is required to support the claim that sfRNA antagonizes FMRP activity. Is it possible to transfect sfRNA (and non-FMRP binding mutant sfRNA) in the cell and show effects on FMRP activity outside of the context of infection?

1b) Figure 5. shows that Δ10 ZIKV is disproportionately enhanced by FMRP KD. While WT virus was enhanced 3.5 fold, the infection rate of the Δ10 ZIKV rose near 5 fold. However, in Figure 3C silencing of FMRP increases the infection rate of the WT about 6 fold. Are these experiments comparable?

1c) For Figure 2A and B, at least one RBP control antibody, and not just IgG, should be included in this RIP to confirm preferential interaction of FMRP with sfRNA over an unrelated RBP. For Figure 2C, FMRP is relatively abundant in testes with a high viral load but the FMRP-RIP signal is low (again an unrelated RBP RIP is an important control) – how do the authors explain this result?

1d) Can the authors exclude that the enrichment for sfRNA is not due to RNA degradation during the IP?

1e) Although FXR2 is a FMRP translational target, FXR2 also binds sfRNA (Figure 1B). In Figure 6, WT ZIKV upregulates FXR2 expression, presumably through FMRP inhibition, so it is important to confirm that siFXR2 does not cause upregulation of the ZIKV NanoLuc reporter in Figure 4.

2) The possibility that FXR2 is relocalizing to stress granules should be tested using a SG marker.

3) Clarification of the mechanism of restriction. Based on the known function of FMRP, the authors focused on viral translation (subsection “FMRP represses ZIKV infection by blocking viral RNA translation”, last paragraph) but it is unclear whether this protein acts directly on the viral RNA.

3a) Does the protein need to bind the 3'UTR or the ORF for this activity? Does it work directly or indirectly? It is possible that the increase infection could be a consequence of alteration of the expression of FMRP targets. This issue should be considered and different possibilities should be at least discussed.

3b) In Figure 4, what are the kinetics of viral RNA replication in FMRP KD cells with the NanoLuc reporter system? This analysis can add useful mechanistic information.

---

## [Author Response]

Essential revisions:1) Further testing of the specificity of the FMRP-sfRNA interaction is needed.1a) The Δ10 mutation produces less sfRNAs but its attenuation can be due to any number of reasons. For example, the 3'UTR is crucial for recruiting pro-viral RBPs that aid in translation and replication; the Δ10 mutation can disrupt this independently of sfRNAs. Thus, the observation that the attenuated virus is rescued to a higher extent upon FMRP depletion could be due to other non sfRNA related mechanism. More direct evidence is required to support the claim that sfRNA antagonizes FMRP activity. Is it possible to transfect sfRNA (and non-FMRP binding mutant sfRNA) in the cell and show effects on FMRP activity outside of the context of infection?

This is an excellent comment and we agree that demonstration of FMRP modulation by sfRNA out of the context of the infection would strengthen our conclusions. To address this we performed electroporation of sfRNA and a mutant version with deletion of xrRNA1 and xrRNA2, regions required for FMRP-sfRNA interaction. Levels of established targets of FMRP (FXR2 and BRD4) were analyzed by flow cytometry. These experiments show that introduction of sfRNA in uninfected cells leads to increased levels of FXR2 and BRD4, supporting a role for ZIKV sfRNA in FMRP inhibition. Data from these experiments are shown in new Figure 7.

1b) Figure 5. shows that Δ10 ZIKV is disproportionately enhanced by FMRP KD. While WT virus was enhanced 3.5 fold, the infection rate of the Δ10 ZIKV rose near 5 fold. However, in Figure 3C silencing of FMRP increases the infection rate of the WT about 6 fold. Are these experiments comparable?

We appreciate this request for clarification. The experiments referred to are not comparable: Figure 3C was performed with the WT Dakar strain and the experiments in Figure 5 used WT Cambodia strain derived from the previously reported infectious clone. A closer comparison could be performed with the Figure 3—figure supplement 1B where the increase in the WT Cambodia strain was 2.5-fold. We emphasize in the revised manuscript that different ZIKV strains exhibited somewhat different sensitivities to FMRP depletion.

1c) For Figure 2A and B, at least one RBP control antibody, and not just IgG, should be included in this RIP to confirm preferential interaction of FMRP with sfRNA over an unrelated RBP. For Figure 2C, FMRP is relatively abundant in testes with a high viral load but the FMRP-RIP signal is low (again an unrelated RBP RIP is an important control) – how do the authors explain this result?

We agree that IP of an unrelated RBP would provide additional evidence for specificity of ZIKV-FMRP interaction. In response to this comment we performed additional RNA immunoprecipitation assays with the polypyrimidine tract-binding protein (PTB) along with northern blot analysis. This revealed that sfRNA strongly prefers interaction with FMRP compared to PTB. These results are included in new Figure 2C. Regarding the experiment with mouse testes shown in Figure 2C we speculate that efficiency of IP from this tissue was relatively inefficient compared to IP from cultured cells. Nevertheless, this experiment showed clear enrichment of ZIKV RNA with FMRP IP.

1d) Can the authors exclude that the enrichment for sfRNA is not due to RNA degradation during the IP?

The reviewers make a good point concerning the possibility of differential RNA decay during the RNA IP reaction. To address this, we analyzed ZIKV genomes and sfRNA in supernatants from RNA IP reactions by Northern blot side-by-side with input RNA. This showed that both viral genomes and sfRNA remain intact during the IP. These data are shown in new Figure 2C.

1e) Although FXR2 is a FMRP translational target, FXR2 also binds sfRNA (Figure 1B). In Figure 6, WT ZIKV upregulates FXR2 expression, presumably through FMRP inhibition, so it is important to confirm that siFXR2 does not cause upregulation of the ZIKV NanoLuc reporter in Figure 4.

We agree it is of interest to determine what effects FXR2 may have on ZIKV infection as well as FXR1. We performed knockdown of these proteins and observed contrasting effects: FXR2 knockdown decreased rate of ZIKV infection, making it unlikely to repress ZIKV translation, while FXR1 depletion slight enhanced infection. These observations are now shown in new Figure 3—figure supplement 2.

2) The possibility that FXR2 is relocalizing to stress granules should be tested using a SG marker.

We made the interesting observation that FXR2 relocalizes to granular structures upon infection with the Δ10, but not WT, ZIKV. While it is possible these are stress granules, we respectfully submit that testing this hypothesis is beyond the scope of our study.

3) Clarification of the mechanism of restriction. Based on the known function of FMRP, the authors focused on viral translation (subsection “FMRP represses ZIKV infection by blocking viral RNA translation”, last paragraph) but it is unclear whether this protein acts directly on the viral RNA.3a) Does the protein need to bind the 3'UTR or the ORF for this activity? Does it work directly or indirectly? It is possible that the increase infection could be a consequence of alteration of the expression of FMRP targets. This issue should be considered and different possibilities should be at least discussed.

The reviewers raise an important point. Our combined data showing binding of FMRP to ZIKV RNA and correlated enhanced infection/translation due to FMRP knockdown strongly suggest that the effect of FMRP is direct. While this is the most parsimonious explanation, our data do not prove that this is the case and it is possible that alteration in expression of FMRP targets contributes to the effect on ZIKV infection. Likewise, our data do not inform on whether binding to the 3’ UTR or ORF restricts virus translation. These possibilities are noted in the revised manuscript Discussion.

3b) In Figure 4, what are the kinetics of viral RNA replication in FMRP KD cells with the NanoLuc reporter system? This analysis can add useful mechanistic information.

The experiments in Figure 4 were performed at a short time point in the presence or absence of the NS5 inhibitor, NITD008, to account for a possible contribution of viral RNA synthesis to luciferase signals. We do not know the kinetics of RNA synthesis with this modified virus but we observed essentially no difference plus or minus NITD008 in the effects of FMRP knockdown on luciferase levels. This lack of difference is the critical point to note. We interpret these data as evidence supporting a role for FMRP in ZIKV translational repression.